# Toward Identifiable Sparse Autoencoders

Walter Nelson [1]    Theofanis Karaletsos [2]    Francesco Locatello [1]

## Abstract

Recently, sparse autoencoders (SAEs) have emerged as an attractive tool for interpreting and interacting with representations in practical neural networks. While it is common empirical folklore, we also show theoretically that SAEs are highly unstable: different training runs are likely to produce different concept dictionaries and sparse codes. We characterize the model properties that hinder the stability of real-world SAEs, and address each of these problems through minimal changes to the architecture and training procedure. Together, these changes yield two versions of an **i**dentifiable SAE (iSAE), a variant of the standard TopK SAE with lower reconstruction error and improved stability. We explain this improvement theoretically by connecting SAEs with traditional dictionary learning approaches, and show that the dictionaries learned in practice satisfy an approximate restricted isometry condition, rendering the corresponding sparse codes in those models near-identifiable.

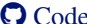 Code

## 1. Introduction

Sparse autoencoders (SAEs) decompose high-dimensional representations into a sparse linear combination of concepts from a large, learned dictionary. Due to their simplicity, flexibility, and well-characterized engineering (Gao et al., 2024), they have seen widespread adoption across vision, language, and other modalities.

In these settings, SAEs are used as a practical interface for analyzing and intervening on the internal representations. For example, individual dictionary atoms are often

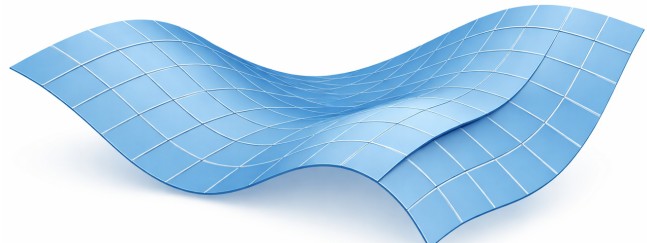

*Figure 1.* Sparse autoencoders approximate nonlinear manifolds (dark blue, mostly occluded) with linear patches (light blue). We show that identifiability hinges on four key ingredients: (a) the approximation being good enough (low reconstruction error), (b) the manifold being sampled densely enough, (c) co-occurring concepts being distinct enough (an approximate restricted isometry property), and (d) sufficiently diverse concept co-occurrence patterns. When these hold, individual patches are identifiable, rendering the whole model statistically identifiable.

interpreted as human-meaningful concepts (e.g., topics, syntactic patterns, or visual components), enabling post-hoc interpretability. They have also been used for steering, where selectively activating or suppressing specific atoms alters model behavior in a targeted way. This line of work frequently relies, implicitly or explicitly, on the assumption that the sparse linear decompositions correspond to stable (or even semantically meaningful) structure in the underlying representation. In this work, we avoid this assumption by treating SAEs not as *a priori* interpretable objects, but as statistical estimators for the classical dictionary learning problem. Our goal is to investigate under what conditions their learned atoms and codes are in fact statistically identifiable.

We are motivated by prior work showing that modern SAEs suffer from various forms of non-identifiability (Song et al., 2025). In particular, prior work has shown that two SAEs trained on the same data are not guaranteed to yield the same concept dictionary (Song et al., 2025) or sparse codes (Paulo & Belrose, 2025), a phenomenon similar to the non-identifiability seen in disentangled representation learning (Locatello et al., 2019). This inconsistency potentially muddies the interpretation of the learned features, defeating the purpose of SAEs in the first place. SAE identifiability is also important when using them to interpret data and models in scientific applications, where identifiability ensures that statistical power is stable (Mencattini et al., 2026;

[1]Institute of Science and Technology Austria [2]Pyramidal Inc. and Achira Inc., USA. Correspondence to: Walter Nelson <walter.nelson@ista.ac.at>.

*Proceedings of the 43rd International Conference on Machine Learning*, Seoul, South Korea. PMLR 306, 2026. Copyright 2026 by the author(s).

Donhauser et al., 2025).

The classical approaches to the compressed sensing and dictionary learning problems faced similar non-identifiability issues, which are mostly due to the fact that the dictionaries learned are highly overcomplete. We draw inspiration from these earlier works to improve the stability of the SAEs widely used in practice today. Concretely:

- We show theoretically (Theorem 3.2) and empirically that consistently learning the dictionary is **not sufficient** for consistently learning the codes, motivating our proposed **code consistency** metrics.

- We show theoretically that when the learned dictionary satisfies an approximate restricted isometry property (aRIP), the sparse codes of an SAE are provably near-identifiable (Theorem 3.6).

- Empirically, we show that two variants of our model identifiable SAE (**iSAE**) exhibit improved stability and reconstruction over standard TopK baselines.

## 2. Related Work

A growing body of recent work applies SAEs as a tool for post-hoc interpretability, including mechanistic interpretability, particularly in large language models (Elhage et al., 2022; Olah et al., 2020; Bricken et al., 2023). In these settings, SAEs are trained on intermediate activations (e.g., residual streams or MLP layers), and individual dictionary elements are interpreted as corresponding to human-meaningful concepts such as semantic topics, syntactic patterns, or behavioral circuits. This approach has enabled analyses of feature superposition (Elhage et al., 2022), circuit structure (Olah et al., 2020), and the localization of model behaviors, as well as interventions in which specific features are ablated or amplified to steer model outputs (Bricken et al., 2023; Turner et al., 2024). These applications rely on the empirical observation that SAE features are often sparse, localized, and partially interpretable, even in highly overcomplete regimes, provided that reconstruction performance is good enough.

Accordingly, most work on SAEs has focused on improving reconstruction error (Gao et al., 2024; Bussmann et al., 2024) at the frontier of the reconstruction-sparsity tradeoff (Fel et al., 2025). However, the evidence is unclear as to whether reconstruction performance is sufficient to give stability of the atoms and sparse codes. Song et al. (2025) point out the importance of the stability of the learned dictionary, arguing that TopK SAEs are stable according to this measure under re-trainings using a standard cosine similarity measure on the concept dictionary. Fel et al. (2025) propose to constrain the concepts in the dictionary to the convex hull of the data, finding that this improves stability according

to the same cosine similarity measure. Building on prior work which shows that two SAEs with the same dictionary can produce vastly different sparse codes for the same data (Paulo & Belrose, 2025), we show that code identifiability is not implied by dictionary identifiability. Accordingly, we instead analyze the stability of both the dictionary and the sparse codes.

In most works on mechanistic interpretability, identifiability is assessed subjectively, by checking the alignment with human-interpretable concepts (Karvonen et al., 2025). In contrast, the historical view in sparse coding and dictionary learning is to assess when signals admit a *stable* sparse linear decomposition, without assigning a particular interpretation to it. Our work aims to bridge this gap by explicitly analyzing the conditions under which the representations learned by SAEs are stable and statistically identifiable, and by proposing metrics that capture this stability at the level of both dictionaries and codes.

Our theoretical characterization is based on the restricted isometry property (RIP) from compressed sensing. Candes et al. (2005) show that when a known dictionary satisfies the RIP, sparse codes are identifiable. Unfortunately, the classical RIP condition is combinatorially difficult to verify, a situation that does not become any easier when the dictionary is unknown (Spielman et al., 2012). To address this, we propose a data-driven relaxation of the RIP condition we refer to as approximate RIP (aRIP), which enforces the property only where data is actually observed. We show theoretically and empirically that this is sufficient for near-identifiability of both the dictionary and the codes for a given data distribution. Appealingly, given a trained SAE, the aRIP condition is easily measured.

## 3. Unifying SAEs and Dictionary Learning

In this section, we contrast sparse autoencoders and the classical approaches to dictionary learning, with the goal of unifying them in theory and practice.

**Sparse autoencoding.** Consider a distribution $P(\mathbf{x})$ over representations $\mathbf{x} \in \mathbb{R}^N$. A typical sparse autoencoder (SAE) takes the form:

$$\mathbf{z} = \mathbf{f}(\mathbf{x}) \qquad \hat{\mathbf{x}} = D\mathbf{z} + \mathbf{b}' \qquad (1)$$

where $\mathbf{f} : \mathbb{R}^N \to \mathbb{R}^K$ is some encoder, $\mathbf{b}' \in \mathbb{R}^N$ is a learned bias term, and $D \in \mathbb{R}^{N \times K}$ is the learned dictionary of $N$-dimensional atoms (or concepts, in SAE parlance). Typically, the encoder is near-linear of the form

$$\mathbf{f}(\mathbf{x}) = \sigma(W(\mathbf{x} - \mathbf{b})) \qquad (2)$$

where $W \in \mathbb{R}^{K \times N}$ is the learned encoder matrix, $\sigma$ is some sparsifier, and $\mathbf{b} \in \mathbb{R}^N$ is a bias term. The parameters are optimized by stochastic gradient descent to minimize

$\|\mathbf{x} - \hat{\mathbf{x}}\|^2$. Generally, the ambient dimension of the code space $K \gg N$, but codes themselves $\mathbf{z}$ are $k$-sparse for some fixed $k \ll K$. The current most popular choice for $\sigma$ is the TopK function (Makhzani & Frey, 2013), which sparsifies to the top $k$ largest coefficients. We refer to this architecture throughout as a TopK SAE, which also sets $\mathbf{b} = \mathbf{b}'$ (Gao et al., 2024).

**Sparse coding and dictionary learning.** In sparse coding, the dictionary $D$ is fixed, and the goal in practice is to solve the optimization problem for a given input $\mathbf{x}$:

$$\min_{\mathbf{z}} \|\mathbf{x} - D\mathbf{z}\|_2 \text{ s.t. } \|\mathbf{z}\|_0 \leq k \qquad (3)$$

Traditional approaches include basis pursuit (Chen & Donoho, 1994) and orthogonal matching pursuit (OMP) (Chen et al., 1989). Optimization is generally intractable due to the highly non-convex nature of the $\ell_0$ norm constraint and the resulting NP-hardness of the problem. Furthermore, even when it can be relaxed to the $\ell_1$ form, without conditions on $D$, the solution to the relaxation of (3) is known to be non-identifiable for general $\mathbf{x}$. This means that multiple sparse codes $\mathbf{z}$, $\mathbf{z}'$ are equally good at representing the same observation $\mathbf{x}$, even when the dictionary is fixed. When the dictionary is learned, the situation is even more complicated. Most algorithms, such as K-SVD (Aharon et al., 2006), alternately solve a relaxation of (3) for numerous samples and update the dictionary.

### 3.1. Toward a unified approach

Sparse autoencoders have several appealing properties. They are easily implemented in modern deep learning frameworks and trained on modern hardware using stochastic gradient descent (Gao et al., 2024; Karvonen et al., 2025; Fel et al., 2025). For large language models in particular, their lightweight architecture means that a large cache of input activations can be inferred while the SAE is trained "online", allowing training to scale to billions of tokens for the largest models (Karvonen et al., 2025; Gao et al., 2024). On the other hand, the solutions that SAEs learn are unstable in practice (Paulo & Belrose, 2025). For example, even if in some settings the dictionaries appear to be relatively stable in practice (Song et al., 2025), the amortized sparse codes themselves often are not (Paulo & Belrose, 2025). This means that the same SAE model trained twice on the same dataset and model might yield different concepts and sparse codes that affect downstream applications of the model.

Dictionary learning has the opposite characteristics. In general, it requires specialized approaches to optimization (Mairal et al., 2009) that fail to scale to the million- or billion-token regime required for SAEs to be useful for interpreting modern models. However, the solutions found by dictionary learning algorithms are often *identifiable* in theory and practice, meaning that the recovered dictionary and codes are learned consistently (Spielman et al., 2012). Given the high-stakes nature of the many common applications of SAEs, such as steering (O'Brien et al., 2025), interpretability (Cunningham et al., 2023) and oversight (Li et al., 2026), identifiability seems like a "bare minimum" requirement, highlighting the potential for bridging the gap between dictionary learning and SAEs.

In mechanistic interpretability applications, identifiability is often framed as recovering the "true" data-generating concepts under the linear representation hypothesis, a form of structural identifiability (Nelson et al., 2026). However, without access to the true concepts, this form of identifiability is impossible to evaluate. However, the run-to-run stability of the learned concepts (or atoms, in the language of dictionary learning) and sparse codes is easily assessed, including on real-world data. As such, we adopt the following definition of statistical identifiability for SAEs.

**Definition 3.1** (SAE Identifiability). Let $P(\mathbf{x})$ denote a data distribution supported on $\mathcal{X}$. Let $\mathbf{f}, \mathbf{f}' : \mathcal{X} \to \mathcal{Z} \subseteq \mathbb{R}^K$ denote the encoders of two $k$-sparse SAEs of the form in (1) trained independently on data from $P(\mathbf{x})$, and let $D, D' \in \mathbb{R}^{N \times K}$ denote their decoders. Then, the SAE model is nearly identifiable in the limit of infinite data from $P(\mathbf{x})$ if there exist $\epsilon_z, \epsilon_D \geq 0$ such that $\|\mathbf{f}(\mathbf{x}) - \Pi\mathbf{f}'(\mathbf{x})\|_2 \leq \varepsilon_z$ almost everywhere and $\|D - D'\Pi\| \leq \epsilon_D$ for some signed permutation matrix $\Pi \in \{-1, 0, 1\}^{K \times K}$.

Intuitively, this definition says that an SAE is identifiable if independent trainings of the model (on infinite data) are guaranteed to yield approximately the same dictionary and sparse codes, up to some trivial equivariances in the model, namely the ordering and sign of the atoms in the dictionary (and therefore the sparse codes). Clearly, this definition is a "pre-requisite" for identifiability of the kind often pursued in mechanistic interpretability (Karvonen et al., 2025): if run-to-run identifiability is poor, identifiability of some ground-truth is necessarily poor as well. The benefit to this definition is that we don't need to assume a particular data-generating process, and identifiability can therefore be assessed empirically with minimal assumptions.

Leveraging recent work on SAEs and a long history of dictionary learning results, we outline the following *"dark triad"* of SAE characteristics that hinder the understanding of identifiability in both theory and practice:

1. **Bidirectional features.** Most activation functions for SAEs, such as ReLU and TopK, restrict the non-zero coefficients of the sparse code $\mathbf{z}$ to be positive. As a result, Zhu et al. (2025) found that the dictionaries of trained SAEs often contain opposing concepts: for example, both a given atom $\mathbf{d}$ and an atom approximating its opposite $\tilde{\mathbf{d}} \approx -\mathbf{d}$ will be present in the dictionary. In practice, this cuts the representation capacity of

the dictionary in half, and renders the applicability of dictionary learning theory unclear at best (note that equation (3) does not specify $\mathbf{z} \geq 0$), because $\mathbf{d}$ and $\tilde{\mathbf{d}}$ are *coherent* (have near-maximal absolute similarity), a known barrier to identifiability (Candes et al., 2005).

2. **Dictionary conditioning.** In general, solutions to equation (3) are unidentified without conditions on the concept dictionary $D$. For example, $D$ must have low mutual coherence (maximum pairwise absolute concept similarity), or satisfy a condition like the restricted isometry property (Candes et al., 2005) in order for the sparse codes $\mathbf{z}$ to be uniquely recoverable from noisy observations. It is poorly understood whether SAEs learn dictionaries for which the corresponding sparse coding problem is identifiable.

3. **Encoder expressiveness.** Even in the simplest case where the dictionary $D$ is fixed and known, it is known that amortized inference of sparse codes is a difficult problem. For example, Gregor & LeCun (2010) shows that a specialized recurrent architecture is required to learn a good mapping $\mathbf{x} \mapsto \mathbf{z}$ from observations to sparse codes, *even when paired observations* $(\mathbf{x}, \mathbf{z})$ *are available from an oracle solver of equation (3)*. This suggests that the simple near-linear encoder typically employed in SAEs described in equation (2) is not expressive enough to learn the usual solutions to the sparse coding problem (3), which are the only solutions for which practical identifiability theory is available.

In the next sections, we describe the changes we make to the usual SAE model (1) that improve the performance of the model and advance our understanding of identifiability in this setting. The goal is to design a model with the "best of both worlds": the well-characterized theory and guarantees of dictionary learning, with the ease of training and scalable engineering of sparse autoencoders.

## 3.2. Bidirectional features

Zhu et al. (2025) propose to change the TopK activation function in SAEs to an absolute value form motivated by the proximal gradient approach to solving equation (3). Specifically, they define:

$$(\text{AbsTopK}_k(u))_i = \begin{cases} u_i, & i \in \mathcal{H}_k(u), \\ 0, & i \notin \mathcal{H}_k(u). \end{cases} \quad (4)$$

where $\mathcal{H}_k(u)$ gives the indices of the $k$ largest coefficients in absolute value. This addresses the first component of the "dark triad", improving the expressivity of the model at the same number of parameters, reducing reconstruction error, and aligning more closely with existing approaches to dictionary learning, which allow for negative coefficients as in the $\ell_0$ and $\ell_1$ solutions to equation (3).

## 3.3. Dictionary conditioning

In statistics, the problem of estimating the dictionary $D$ in this setting has a long history under the name "sparse overcomplete dictionary learning", and has known estimation challenges. Indeed, even when $D$ is known, the codes $\mathbf{z}$ might be non-identifiable (Candes et al., 2005), meaning multiple distinct sparse decompositions might exist. The issue becomes even worse under imperfect reconstruction, which is observed in practice in the SAE setting. To illustrate this failure mode, we construct a generic dictionary which can fail to identify codes, even if it approximates a reasonable observational distribution well in reconstruction. This is formalized in the following impossibility theorem.

**Theorem 3.2** (Identifiability Impossibility). *There exists a normalized dictionary $D \in \mathbb{R}^{N \times K}$ with the following property. Let $\mathbf{f} : \mathcal{X} \to \mathbb{R}^K$ be any $k$-sparse continuous encoder defining an SAE $(f, D)$ with reconstruction error tolerance $\epsilon \geq 0$ such that $\|D\mathbf{f}(x) - x\| \leq \epsilon$ for almost all $x$ drawn from a distribution $P(x)$. Assume further that every concept in $D$ is activated with positive probability under $P(x)$. Then, there exists another continuous encoder $\mathbf{f}' : \mathcal{X} \to \mathbb{R}^K$ that achieves the same reconstruction accuracy, but differs from $\mathbf{f}$ on a set of inputs with positive probability. Moreover, the sparsity patterns of $\mathbf{f}(x)$ and $\mathbf{f}'(x)$ differ with positive probability.*

The proof is given in Appendix A. Theorem 3.2 shows that even if an SAE learns a perfectly stable dictionary, it may be the case that the corresponding sparse coding problem is non-identifiable. Indeed, the dictionary learning literature shows that if the dictionary is not fixed in advance but instead learned, the potential sources of non-identifiability are even more numerous (Spielman et al., 2012).

We now ask what conditions we can place on the SAE model in (1) to render both the dictionary and the codes identifiable, or nearly so. (Candes et al., 2005) show that a restricted isometry property (RIP) condition on the dictionary $D$ is sufficient to render the codes unique, up to permutations. The condition takes the form

$$(1 - \delta)\|\mathbf{z}\|^2 \leq \|D\mathbf{z}\|^2 \leq (1 + \delta)\|\mathbf{z}\|^2 \quad (5)$$

for some $\delta \geq 0$, and must hold for any $k$-sparse $\mathbf{z}$. Intuitively, it says that the reconstructions must have approximately the same norms as the corresponding sparse codes. If $D$ were square, any nearly orthogonal matrix would satisfy the condition. But, when $D$ is overcomplete and is learned, the RIP condition is combinatorially difficult to enforce, because every dictionary has $K$ choose $k$ possible combinations of concepts where the condition must be checked. Intuitively, for overcomplete dictionaries, any combination of $k$ atoms must be "nearly orthogonal".

To address this difficulty of combinatorial complexity, we propose to enforce (5) only on the sparse codes $\mathbf{z}$ actually

observed in the latent space of the sparse autoencoder. We refer to this condition as an approximate restricted isometry property, because it of course does not imply the usual RIP condition. However, we will show that it is sufficient to prove useful identifiability results.

**Definition 3.3** (Approximate RIP). Let $P(\mathbf{z})$ be a distribution over sparse codes, and let $\mathbf{z}_1, \mathbf{z}_2$ be independent samples from $P(\mathbf{z})$. Define $S$ to be a random variable, representing the union of the indices where $\mathbf{z}_1$ and $\mathbf{z}_2$ are non-zero. Then, $D$ satisfies the **approximate restricted isometry property** (aRIP) with respect to $P(\mathbf{z})$ at level $\delta$ if $(1 - \delta)\|\mathbf{z}'_S\|^2 \leq \|D_S \mathbf{z}'_S\|^2 \leq (1 + \delta)\|\mathbf{z}'_S\|^2$ for any vector $\mathbf{z}'_S \in \mathbb{R}^K$ such that its nonzero coefficients have indices $S$, almost everywhere with respect to $P(S)$.

Intuitively, this means that any union of two *observed* sparsity patterns in the sparse code distribution must satisfy the usual restricted isometry property (RIP). This is in contrast to more general definitions which demand the condition hold for the union of *any* two (or three) sparsity patterns to achieve identifiability. Such definitions can be used to prove identifiability results that hold for all possible observable signals $\mathbf{x} \in \mathbb{R}^N$ (Candes et al., 2005). In contrast, ours will only hold for a particular distribution of signals $P(\mathbf{x})$. For these results to hold, we will need the following assumptions on the distribution $P(\mathbf{z})$, which in the case of a sparse autoencoder of the form in (1) is the pushforward of the data distribution $P(\mathbf{x})$ by the encoder $f$.

**Assumption 3.4** (Sufficient Richness). Let $\mathbf{z}$ and $\mathbf{z}'$ be independent sparse codes from $P(\mathbf{z})$, with nonzero index sets $S$ and $S'$ respectively. Then $P(\mathbf{z})$ satisfies **sufficient richness of supports** if $P(S \cap S' = \{i\}) > 0$ and $P(i \in S, j \in S', S \cap S' = \emptyset) > 0$ for any pair of atoms $i \neq j$. Intuitively, these imply that no atom can occur in a single support $S$ and no two atoms can always co-occur.

**Assumption 3.5** (Sufficient Diversity). Let $\mathbf{z}$ and $S$ be as defined in Assumption 3.4. Then, $P(\mathbf{z})$ satisfies **sufficient diversity of observations** if for each observed support $S$ of size $k$, there exist $k$ independent samples from $\mathbf{z}_S \mid S$ such that the $k \times k$ matrix formed by stacking these samples has smallest singular value at least $\zeta > 0$.

**Measuring aRIP**   Because we don't need to sample every possible combination of sparsity patterns to quantify aRIP (Definition 3.3), we might reasonably hope to measure how well the condition holds for a given distribution of sparse codes $P(\mathbf{z})$ using samples from the distribution. The key algebraic relationship to notice is that if $\lambda_1, \ldots, \lambda_u$ are the eigenvalues of the Gram submatrix $G_S = D_S^\mathsf{T} D_S$ in ascending order, the optimal aRIP constant $\delta$ satisfies $1 - \delta \leq \lambda_1 \leq \lambda_s \leq 1 + \delta$. When $D$ is normalized, the mean

eigenvalue $\bar{\lambda} = 1$, and we have the following relationship:

$$\mathcal{R}_{\text{aRIP}}(S) := \frac{1}{|S|} \sum_{i=1}^{|S|} (\lambda_i - 1)^2 = \frac{\text{tr}(G_S^2)}{|S|} - 1 \quad (6)$$

Intuitively, this equation measures the *average* deviation of each eigenvalue from 1, whereas the aRIP condition witnesses only the *largest* deviations of the eigenvalues from 1. However, the eigenvalue bound does imply that at the minimum of $\mathcal{R}_{\text{aRIP}}(S)$ over the space of normalized subdictionaries $D_S$, we have $\delta = 0$.

We draw inspiration from work which regularizes the input-output Jacobian of neural networks for improving representation learning (Lee et al., 2022), allowing us to straightforwardly estimate (6). Let $S$ be a set of atom indices. With a slight abuse of notation due to the random dimensionality of $S$, let $\mathbf{n}_S \sim \mathcal{N}(0, \mathbf{I}_{|S|})$, so we have (up to an additive constant):

$$\mathbb{E}_S[\mathcal{R}_{\text{aRIP}}(S)] = \mathbb{E}_S \left[ \mathbb{E}_{\mathbf{n}_S} \left[ \frac{\|D_S^\mathsf{T} D_S \mathbf{n}_S\|^2}{|S|} \right] \right] \quad (7)$$

Ideally, we would compute (7) for all possible input pairs $\mathbf{x}_1, \mathbf{x}_2$ by setting $S = S_1 \cup S_2$ where $S_i$ is the set of nonzero indices for the sparse code $\mathbf{z}_i = \mathbf{f}(\mathbf{x}_i)$. However, this would be prohibitively expensive, so we make some approximations for computational tractability. Intuitively, for index sets of identical size, if their spans $D_{S_1}$ and $D_{S_2}$ are orthogonal to one another, then $\mathcal{R}_{\text{aRIP}}(S)$ is equal to the sum of the individual terms $\mathcal{R}_{\text{aRIP}}(S_1)$ and $\mathcal{R}_{\text{aRIP}}(S_2)$. On the other hand, if the spans are oblique to one another, $\mathcal{R}_{\text{aRIP}}(S)$ depends on the interaction between the two sparse supports, and can be much larger than the sum of the two.

Motivated by this, we leverage the amortized encoder $\mathbf{f}$ in (1) to determine the sets $S$ to quantify. In particular, given the unsparsified code $\tilde{\mathbf{z}}_i \in \mathbb{R}^K$ for a given input $\mathbf{x}_i$, we define $S$ to be the indices of the largest $2k$ coefficients of $\tilde{\mathbf{z}}$ in magnitude, where $k$ is the sparsity level. Because the encoder (2) is near-linear, this tends to have the effect of "picking" a sparse index set $T$ of size $k$ such that its span is maximally parallel to that of $S_i$, and setting $S = S_i \cup T$. This ensures we select "maximally interactive" pairs of supports, which are most likely to be problematic.

Given the relatively low sparsity levels $k$ we're interested in, a single sample is enough to estimate the expectation (7). Indeed, given the simplicity of equation (7), we will see that we can even use $\mathcal{R}_{\text{aRIP}}$ as a regularizer to obtain dictionaries that better satisfy the approximate RIP condition. The following theorem shows that this leaves us with a form of near-identifiability of the dictionary and sparse codes of the SAE, under mild additional assumptions on the sparse code distribution, whenever reconstruction error is low.

**Theorem 3.6.** *Consider two SAEs of the form (1), optimized with $\mathcal{L}_\theta(\mathbf{x}) = \|\mathbf{x} - \hat{\mathbf{x}}\|^2$ such that they achieve reconstruc-*

*tion error $\mathcal{L}_\theta(\mathbf{x}) \le \epsilon$ almost everywhere on $\mathcal{X}$, satisfying aRIP (Definition 3.3) at level $\delta$ with respect to the pushforward of the data distribution by the encoder, and satisfying Assumptions 3.4 and 3.5. Then, if the dictionaries of the two SAEs are $D$ and $D'$ respectively with codes $\mathbf{z}$ and $\mathbf{z}'$, we have*

$$\|D - D'\Pi\| \le \epsilon_D \tag{8}$$
$$\|\mathbf{z} - \Pi\mathbf{z}'\| \le \epsilon_z \tag{9}$$

*for some signed permutation matrix $\Pi \in \{-1, 0, 1\}^{K \times K}$ and error terms $\epsilon_D$ and $\epsilon_z$ which are functions of $\epsilon$, $\rho$, $k$ and properties of the data distribution $P(\mathbf{x})$, where the second inequality holds almost surely. Furthermore, $\epsilon_D, \epsilon_z \xrightarrow{\epsilon, \rho \to 0} 0$.*

Our theorem shows that if an SAE yields dictionaries which satisfy the aRIP condition on unions of sparse supports from independent pairs of samples, its sparse codes are nearly identifiable up to permutations, meaning the same sparse codes (and therefore the same dictionaries) have been recovered. The level of nearness is governed by the reconstruction error and the level of aRIP regularization achieved by the solutions. The proof is given in Appendix A.

### 3.4. Encoder expressiveness

The identifiability theory in the previous section assumes an encoder that can yield sparse codes with low reconstruction error for the given dictionary. Importantly, this means the encoder must be *sufficiently expressive* to capture the mapping $\mathbf{x} \mapsto \mathbf{z}$ of observations to sparse codes. However, existing architectures for SAEs use a near-linear encoder, with the only source of non-linearity being a sparsifier. Early work on neural sparse coding (Gregor & LeCun, 2010) found such encoders to be insufficiently expressive, even when the dictionary is fixed, an observation also noted with SAEs (Donhauser et al., 2025). Gregor & LeCun (2010) propose a multistep encoder architecture motivated by the iterative soft thresholding algorithm to resolve this. In our implementation, each step of the encoder takes the form:

$$\mathbf{z}^{(i)} = \text{AbsTopK}_k(S\mathbf{z}^{(i-1)} + W(\mathbf{x} - \mathbf{b})) \tag{10}$$

where $\mathbf{z}^{(0)} = \mathbf{0}$, so the only additional parameter is the (learned) step size matrix $S$. We replace the near-linear encoder from equation (2) with $\mathbf{f}(\mathbf{x}) = \mathbf{z}^{(T)}$, where $T$ is the number of iterations. In practice, we use $T = 5$ in all experiments, although the results don't seem particularly sensitive to choices between 3 and 10.

## 4. Experiments

In our experiments, we evaluate the role of bidirectional features (section 3.2), dictionary conditioning (section 3.3)

*Table 1.* Performance and identifiability metrics on **synthetic** data. Model variants shown are: TopK; AbsTopK (bidirectional features); iSAE (bidirectional features + aRIP regularization); and iSAE-ME (bidirectional features + aRIP regularization + multistep encoding). MSE = mean squared reconstruction error, $\mathcal{R}_S$ = aRIP measurement, DCS = dictionary cosine similarity, IoU = intersection over union, $\ell_2 = \ell_2$ error.

| Model | MSE | $\mathcal{R}_S$ | Pairwise Identifiability | | | | DCS |
|---|---|---|---|---|---|---|---|
| | | | SAE | | Oracle | | |
| | | | IoU | $\ell_2$ | IoU | $\ell_2$ | |
| *i.i.d.* | | | | | | | |
| TopK | 0.447 | 0.098 | 0.113 | 0.882 | 0.172 | 0.786 | 0.606 |
| AbsTopK | 0.318 | 0.106 | 0.487 | 0.557 | 0.914 | 0.176 | 0.935 |
| **iSAE** | 0.318 | 0.101 | 0.487 | 0.558 | 0.913 | 0.177 | 0.933 |
| **iSAE-ME** | **0.001** | 0.102 | **0.997** | **0.020** | 0.997 | 0.012 | **0.999** |
| *mixture* | | | | | | | |
| TopK | 0.357 | 0.120 | 0.149 | 0.840 | 0.250 | 0.758 | 0.633 |
| AbsTopK | 0.303 | 0.229 | 0.393 | 0.637 | 0.675 | 0.427 | 0.804 |
| **iSAE** | 0.319 | 0.095 | 0.497 | 0.526 | 0.706 | 0.389 | 0.843 |
| **iSAE-ME** | **0.040** | 0.230 | **0.873** | **0.219** | 0.877 | 0.215 | **0.932** |

and encoder expressiveness (section 3.4) in the performance of the SAE. The main goal is to understand the impact of dictionary conditioning and design decisions on reconstruction performance and the level of empirical identifiability. To assess empirical identifiability, we fit multiple SAEs with identical hyperparameters to the same data but with different seeds for initialization and training.

**Metrics** We measure performance by the usual mean squared error of the reconstructions $\|\mathbf{x} - \hat{\mathbf{x}}\|^2$. For each pair of SAEs, identifiability is measured along two axes: dictionary identifiability and code identifiability. We match the concepts in the pair of learned dictionaries according to their cosine similarity via the Hungarian algorithm (Song et al., 2025). The mean absolute dictionary cosine similarity (DCS) after matching is reported. The same matching is used to align the sparse codes $\mathbf{z}$ (taking care to align signs as well), and the intersection-over-union (IoU) in the matched sparsity patterns, along with the $\ell_2$ error (normalized by the mean norm of the codes $\mathbf{z}$) is reported.

**Sparse coding oracle** As we are also interested in studying the relationship between dictionary conditioning, dictionary identifiability, and sparse code identifiability, we also explore the use of an *oracle solver* to obtain sparse codes from the learned dictionary. Specifically, we apply orthogonal matching pursuit (OMP; Pati et al. (1993)) to the learned dictionary from each of the SAEs, and assess whether it recovers the same sparse codes by employing the same identifiability metrics described above (IoU and $\ell_2$ error).

**Settings** We consider three settings. The first two are synthetic. We generate 768-dimensional data noiselessly

from a "true" dictionary consisting of 4096 concepts, which are unit vectors generated uniformly either at random (the i.i.d. case) or from a mixture of Gaussians projected onto the unit sphere (the mixture case). The codes are simulated as i.i.d. unit Gaussian random variables, sparsified to sparsity level $k$ by zeroing out all but the $k$ largest elements in magnitude. To assess the performance of our models in the real world, we evaluate on activations from layer 12 of Pythia-160M (Biderman et al., 2023) on The Pile (Gao et al., 2020). These activations are 768-dimensional, and we train SAEs with a dictionary size of 4096 concepts and a sparsity level of $k = 40$, the same size as our synthetic experiments. We also assess the models on patch tokens from DINOv2-Base (Oquab et al., 2024) using ImageNet-1k (Deng et al., 2009). These tokens and the SAE have the same dimensionality (768-dimensional activations, 4096 concepts, $k = 40$).

**Training recipe** During the course of our experiments, we found that both the performance and stability of SAEs are highly dependent on the training recipe. We use the standard training recipe from Gao et al. (2024) for all of our experiments, detailed here:

- **Normalization.** The inputs and dictionary atoms (concepts) are normalized to have unit norm.

- **Tied intercept.** The pre- and post-bias in equation (1) are tied to be the same parameter, i.e. $\mathbf{b} = \mathbf{b}'$.

- **Initialization.** The intercept is initialized to the geometric median of the data. The encoder is initialized to the approximate left inverse of the decoder.

- **Auxiliary loss.** We employ an auxiliary loss to prevent dead concepts.

In all experiments, these components of the model and training procedure are exactly the same for all models compared, including the TopK SAE baseline. Furthermore, all models are always trained for the same number of training steps (512M total tokens for the LLM model in batches of 2048, 40K steps for the synthetic model with a batch size of 2048). The synthetic, LLM and vision training schemes are all "online", in the sense that no tokens are repeated during training. For LLM and vision training in particular, a buffer of 500K tokens is inferred, and batches are drawn at random from this buffer, which is replenished when it's half-empty.

### 4.1. Bidirectional features

As described in section 3.2, we implement the AbsTopK activation function as an alternative to the more standard TopK activation function. This allows negative concept loadings in the sparse codes, and effectively doubles the capacity of the dictionary at the same number of parameters.

*Table 2.* Performance and identifiability metrics on **LLM & vision** activations. Model variants shown are: TopK; AbsTopK (bidirectional features); iSAE (bidirectional features + aRIP regularization); and iSAE-ME (bidirectional features + aRIP regularization + multistep encoding). MSE = mean squared reconstruction error, $\mathcal{R}_S$ = aRIP measurement, DCS = dictionary cosine similarity, IoU = intersection over union, $\ell_2 = \ell_2$ error.

| Model | MSE | $\mathcal{R}_S$ | Pairwise Identifiability | | | | DCS |
|---|---|---|---|---|---|---|---|
| | | | SAE | | Oracle | | |
| | | | IoU | $\ell_2$ | IoU | $\ell_2$ | |
| *Pythia-160M* | | | | | | | |
| TopK | 0.166 | 0.183 | 0.398 | 0.492 | 0.350 | 0.530 | 0.813 |
| AbsTopK | 0.177 | 0.183 | **0.631** | **0.212** | 0.398 | 0.292 | **0.873** |
| **iSAE** | 0.179 | 0.114 | 0.526 | 0.256 | 0.472 | 0.273 | 0.858 |
| **iSAE-ME** | **0.148** | 0.132 | 0.375 | 0.297 | 0.343 | 0.316 | 0.797 |
| *DINOv2* | | | | | | | |
| TopK | 0.217 | 0.163 | 0.519 | 0.442 | 0.320 | 0.576 | 0.836 |
| AbsTopK | 0.235 | 0.159 | 0.524 | 0.424 | 0.374 | 0.462 | **0.862** |
| **iSAE** | 0.237 | 0.130 | **0.585** | **0.372** | 0.386 | 0.449 | 0.846 |
| **iSAE-ME** | **0.191** | 0.137 | 0.356 | 0.451 | 0.305 | 0.494 | 0.807 |

**Results: synthetic** In both the i.i.d. and mixture synthetic cases (Table 1), transitioning TopK → AbsTopK substantially improves reconstruction error, by around 30-40%. With this improvement in reconstruction comes a substantial improvement in the identifiability of the dictionary, from essentially unidentified (a DCS of 0.6 corresponds to an average angle of over 45 degrees between concept loadings, which is significant in 768-dimensional space) to well but not perfectly identified. Importantly, these gains in dictionary identifiability *can* be realized by a good sparse coding algorithm: the OMP oracle achieves excellent identifiability using this dictionary, even in the mixture case which is clearly more challenging due to the coherence in the data-generating dictionary. However, the usual encoder seemingly cannot realize these gains, because the SAE has much worse code identifiability than the oracle.

**Results: LLM & vision** The results on Pythia-160M and DINOv2 activations are similar to one another, but different from the results in synthetic data. AbsTopK makes for worse reconstruction error than TopK in both models, a result which is not consistent with prior work on AbsTopK, reflecting potential differences in the training procedure (Zhu et al., 2025). However, in both settings AbsTopK models do have the highest dictionary stability and correspondingly the oracle identifiability metrics are improved relative to TopK. Unlike in the synthetic regime, however, the SAE is better able to exploit dictionary stability for improved code stability, with higher IoU and better $\ell_2$ error than the oracle. This suggests that dictionary stability as measured by DCS is at least partly orthogonal to dictionary conditioning, because the oracle should always be able to exploit a (globally) well-conditioned dictionary at least as well as the SAE.

*Table 3.* Downstream performance of the resulting SAEs trained on activations from Pythia-160M, as evaluated by SAEBench. CE Loss measures how well the SAE reproduces activations such that the LLM's loss is not excessively inflated (lower is better). Sparse Probing accuracy measures how well the SAE recovers pre-specified concepts (higher better). Spurious correlation removal (SCR) tests whether spurious correlations can be removed from a downstream supervised probe by zeroing the confounding latent. Targeted probe perturbation (TPP) assesses selectivity in the concepts by zeroing a latent and assessing the impact on other supervised probes.

| Model | CE Loss | Sparse Probing | SCR | TPP |
|---|---|---|---|---|
| TopK | 3.974 | **0.909** | **0.381** | 0.043 |
| AbsTopK | 4.004 | 0.906 | 0.330 | 0.031 |
| **iSAE** | 4.003 | 0.907 | 0.348 | 0.043 |
| **iSAE-ME** | **3.906** | 0.908 | 0.293 | **0.044** |

## 4.2. Dictionary conditioning

Next, we explore the role of dictionary conditioning, by considering the addition of the regularization term (7) to the model (AbsTopK → iSAE). For all experiments using the regularization term (including in the subsequent section), we fix the weight of the term to $10^{-1}$.

**Results: synthetic** In synthetic data, dictionaries learned using the standard training recipe with the AbsTopK activation function as in the previous section satisfy the aRIP condition fairly well (Table 1). Furthermore, the dictionaries are stable, and the corresponding oracle identifiability is good, particularly in the i.i.d. case. In the mixture case, the addition of aRIP regularization is effective in improving the estimated aRIP constant of the dictionary, and confers a marginal improvement in the code identifiability of the oracle but not the SAE.

**Results: LLM & vision** When trained on LLM activations, the results are much clearer. Identifiability of the oracle is improved substantially, with substantial gains in code IoU and improvements in $\ell_2$ error. This occurs despite the slight drop in dictionary stability in both settings, suggesting that dictionary conditioning is the mechanism by which this improved stability occurs. However, these gains are only realized by the linear encoder in DINOv2, and the identifiability gains of the SAE are marginal.

## 4.3. Encoder expressiveness

Finally, we explore the multi-step encoding scheme proposed by Gregor & LeCun (2010) described in section 3.4 (iSAE → iSAE-ME). This introduces a step size parameter $S$ as shown in equation (10), but otherwise does not modify the model.

**Results: synthetic** In the synthetic regime, adding the multistep encoder is crucial to realizing the identifiability gains from the improved dictionary conditioning in the SAE. In particular, even when bidirectional features and dictionary conditioning (previous sections) are enough to improve the quality of the dictionary and therefore the oracle solver, the default near-linear encoder cannot properly amortize codes near the true solution. On the other hand, iSAE-ME learns a nearly perfectly stable model in the synthetic regime as a result of its improved expressivity.

**Results: LLM & vision** On the other hand, in Pythia-160M and DINOv2 activations, the expressive encoder actually worsens identifiability of both the codes and the dictionary. This is in spite of the fact that it attains the **lowest reconstruction error on LLM activations we report in this paper**, with a substantial improvement over all models including baselines. This combination of facts suggests that the primary remaining barrier to identifiability for iSAE-ME is one of optimization, and is not a fundamental limitation of the expressivity of the encoder architecture.

## 4.4. SAEBench: Pythia-160M

When SAEs are used to interact with LLM activations, reconstruction performance and statistical identifiability are only proxies for desirable behaviours such as correctly identifying human-interpretable concepts (Karvonen et al., 2025). We evaluate the two forms of identifiable SAEs (iSAE and iSAE-ME) and compare them to TopK and AbsTopK baselines in terms of how well they minimize impact on the LLM loss, sparse probing performance, and targeted ablation performance (Table 3). iSAE-ME has the best performance in terms of the cross-entropy loss, consistent with its superior reconstruction performance (Table 2). On the other hand, the TopK baseline has the best sparse probing performance, although all models perform reasonably well and the difference appears marginal. Spurious correlation removal (SCR) ablates concepts by setting them to zero, which perhaps biases this task in favour of TopK models where zero has a clear interpretation as "absence" of a concept. Indeed, the TopK baseline performs well on this task, although iSAE-ME outperforms on target probe perturbation (TPP), which checks whether this ablation hampers the performance of other supervised probes. iSAE consistently outperforms AbsTopK, despite its marginally worse reconstruction performance and identifiability in Pythia-160M, suggesting adherence to the aRIP condition is perhaps related to downstream task performance.

## 5. Discussion

In this paper, we have defined a theoretical notion of statistical identifiability that applies to sparse autoencoders.

Specifically, a sparse autoencoder is near-identifiable if its training procedure is "stable": trained on the same distribution, it should yield (nearly) the same concept dictionary and amortized sparse codes, every time. Further, we have assessed whether the commonly used TopK model is identifiable according to this definition, finding that it is not, and showing that improvements to the model in the form of architecture and regularization can greatly improve the near-identifiability and performance of the model. As a result, we present two variants of an SAE that are more identifiable on synthetic and some real-world activations, including one variant which achieves a massive reduction in reconstruction error due to the improved expressivity of its encoder.

SAEs are increasingly being used to interpret and interact with the large-scale neural networks used in practice, such as large language models. In this setting, identifiability of the sparse autoencoder is a "bare minimum" requirement for certain applications. For example, mechanistic interpretability (MI) aims to uncover the "true" hidden workings of the model, assuming that there is a "true mechanism". So, if two trained SAEs uncover two different candidate mechanisms in the form of concepts, they surely cannot both be correct. Furthermore, many approaches for model oversight (Li et al., 2026) and model steering (O'Brien et al., 2025) rely on concept identification, rendering SAE identifiability paramount (Cywiński & Deja, 2025).

Our experiments highlight that quantifying identifiability empirically in this setting is not trivial. In particular, we are the first to directly measure the stability of the sparse codes, proposing to do so using the intersection-over-union metrics on the sparsity patterns and the $\ell_2$ error after aligning using the dictionary. We show that dictionary identifiability, as measured by cosine similarity (Song et al., 2025), is not sufficient to characterize identifiability of the sparse codes. We attribute this to the conditioning of the dictionary: even if the dictionary can be learned stably, this does not render the corresponding sparse coding problem identifiable, as made clear in our impossibility result Theorem 3.2.

Our theory then aims to measure and optimize for dictionary conditioning, in the hopes of improving the identifiability of the sparse codes. To do this, we relax the restricted isometry property (RIP) condition from sparse coding and dictionary learning (Candes et al., 2005) to an approximate form, rendering the condition measurable. Our identifiability result Theorem 3.6 leverages this approximate RIP condition to prove near-identifiability of the dictionary and sparse codes. We emphasize that this of course does not break the NP-hardness of checking the RIP condition on the dictionary, and therefore we cannot claim that SAEs satisfying our aRIP condition correctly recover a particular ground-truth data-generating process, nor that their identifiability properties will generalize outside of the distribution they're trained on.

Experiments show marked benefit to improving the dictionary conditioning. We employed orthogonal matching pursuit (OMP) as an oracle solver just to show that sometimes, the amortized encoder is the bottleneck to learning good sparse codes, even when the dictionary is well-conditioned and learned fairly stably. In the synthetic regime, using a more expressive encoder allows the amortized encoder of the SAE to "catch up" to the oracle. On the other hand, in real-world activations, the oracle is uniformly weaker than the SAE encoder, suggesting that optimization dynamics when training SAEs on real activations are substantially more complex. Notably, the more expressive encoder does allow for massive gains in reconstruction performance, and so we find it interesting to report here.

Importantly, all the modifications that constitute our model scale extremely well. We successfully trained both iSAE and iSAE-ME to learn concept dictionaries of size 4K (4096) on real-world LLM activations (dimension 768). Training times are reasonable: the proposed aRIP regularization term adds negligible overhead, while the multistep encoder increases training times by about 10-15% over a standard TopK SAE. It takes about 5 hours to train iSAE-ME on 512M activations from Pythia-160M, including inference time of the LLM, compared to about 4 hours for the TopK baseline.

**Limitations** Although we are not the first to propose evaluating SAEs and their stability in the synthetic regime (Song et al., 2025), our experiments highlight that there remain gaps between synthetic data-generating processes and real-world activations from practical neural networks. It is of interest to develop more realistic synthetic data-generating processes that facilitate faster model development and the study of failure modes of these models. Furthermore, we experimented only with TopK SAEs. Given that the multistep encoding scheme we adapted from LISTA (Gregor & LeCun, 2010) is built for ReLU-style activation functions, it would be interesting to see how it impacts identifiability and performance in ReLU SAEs as well. This is of particular interest given the conflicting results of the performance of the AbsTopK activation function we present in Section 3.2.

Finally, although our experiments show a massive improvement in code stability, it is still far from perfect in the real settings we consider here. We hypothesize that the remaining gap is largely due to optimization, although we emphasize that despite the large size of the models we train in this paper, it is also possible that they lack the capacity to adequately represent the observation distribution. As noted in Song et al. (2025), this can be a barrier to identifiability, and therefore a larger-scale study of identifiability across model sizes might be warranted.

# Acknowledgements

This work was supported by the Chan Zuckerberg Initiative (CZI) through the AI Residency Program. We thank CZI for the opportunity to participate in this program and the CZI AI Infrastructure Team for support with the GPU cluster used to train our models.

# Impact Statement

The primary application of our improvements to sparse autoencoders is to render highly "black-box" models more interpretable, generally regarded as a good property for responsible use of artificial intelligence. However, interpretability is remarkably difficult to evaluate in practice, and even when achieved does not automatically lead to more responsible use. Accordingly, care must be taken to ensure that users of SAEs do not overstate their reliability or epistemic capability.

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

# A. Proofs

**Notation**  We define an SAE as an encoder-decoder pair $(f, D)$ where $D \in \mathbb{R}^{N \times K}$ is a dictionary with unit-norm columns. For subspaces $\mathcal{U}, \mathcal{V} \subset \mathbb{R}^N$, we denote the minimum and maximum principal angles between them $\theta_{\min}(\mathcal{U}, \mathcal{V})$ and $\theta_{\max}(\mathcal{U}, \mathcal{V})$ respectively. For the sparse support of a code $\mathbf{z}$, $S \subset [n]$, we denote the local RIP constant $\delta_S = \|D_S^T D_S - I\|_{\mathrm{op}}$, also a random variable.

We begin by proving our impossibility result, Theorem 3.2. The key idea is that there exist dictionaries which conceivably could approximate a particular distribution $P(\mathbf{x})$ well, but do not uniquely identify the corresponding sparse codes. We emphasize that this is a well-known result in sparse coding which in fact motivates much of that literature (Candes et al., 2005). For ease of reading, the formal statement of the theorem wordlessly converts the "every concept is activated with positive probability" assumption to an assumption that two explicitly constructed problematic concepts are activated with positive probability.

*Theorem* (3.2, formal). Fix integers $N, K$ and a sparsity level $k$ such that $2 \leq k \leq \min\{N, K - 2\}$. There exists a normalized dictionary $D \in \mathbb{R}^{N \times K}$ and two supports $S, S' \subset [K]$ with $|S| = |S'| = k$ and $|S \cap S'| = k - 2$ such that the following holds.

Let $\mathcal{X}$ be the support of an observation distribution $P(\mathbf{x})$. Let $f : \mathcal{X} \to \mathbb{R}^K$ be a continuous encoder such that $f(\mathbf{x})$ has at most $k$ nonzeros for every $\mathbf{x} \in \mathcal{X}$ and

$$\|Df(\mathbf{x}) - \mathbf{x}\| \leq \epsilon$$

for every $\mathbf{x} \in \mathcal{X}$. Then there exists another sparse autoencoder $(f', D)$ with a continuous encoder $f' : \mathcal{X} \to \mathbb{R}^K$ such that $f'(\mathbf{x})$ has at most $k$ nonzeros for every $\mathbf{x} \in \mathcal{X}$ and

$$\|Df'(\mathbf{x}) - \mathbf{x}\| \leq \epsilon$$

for every $\mathbf{x} \in \mathcal{X}$. Further, if there exists $\mathbf{x}_0 \in \mathcal{X}$ such that $Df(\mathbf{x}_0)$ has a nonzero component along $\mathrm{span}\{e_{k-1}, e_k\}$, then there exists $\mathbf{x}_1 \in \mathcal{X}$ such that $f(\mathbf{x}_1) \neq f'(\mathbf{x}_1)$ and the sparse supports of $f(\mathbf{x}_1)$ and $f'(\mathbf{x}_1)$ are different.

*Proof.* Fix $\varepsilon > 0$. Let $e_1, \ldots, e_N$ denote the standard basis.

Define the dictionary $D$ by specifying its columns. For $1 \leq j \leq k$ set $D_j = e_j$. Define two additional unit vectors in the plane spanned by $e_{k-1}$ and $e_k$:

$$D_{k+1} = \frac{e_{k-1} + \varepsilon e_k}{\sqrt{1 + \varepsilon^2}}$$

and

$$D_{k+2} = \frac{\varepsilon e_{k-1} + e_k}{\sqrt{1 + \varepsilon^2}}.$$

For $j \in \{k+3, \ldots, K\}$ choose any unit vectors $D_j$ in $\mathrm{span}\{e_1, \ldots, e_k\}$. Every column has unit norm, so $D$ is normalized.

Define the two supports

$$S = \{1, 2, \ldots, k\}$$

and

$$S' = \{1, 2, \ldots, k-2, k+1, k+2\}.$$

Then $|S| = |S'| = k$ and $|S \cap S'| = k - 2$.

Denote the $k$-dimensional subspace by $\mathcal{U} = \mathrm{span}\{e_1, \ldots, e_k\}$. Since every column of $D$ lies in $\mathcal{U}$, we have $D\mathbf{z} \in \mathcal{U}$ for every $\mathbf{z} \in \mathbb{R}^K$.

**Claim.** There exist continuous maps $g_S : \mathcal{U} \to \mathbb{R}^K$ and $g_{S'} : \mathcal{U} \to \mathbb{R}^K$ such that for every $\mathbf{u} \in \mathcal{U}$, the vector $g_S(\mathbf{u})$ is supported on $S$, the vector $g_{S'}(\mathbf{u})$ is supported on $S'$, and

$$Dg_S(\mathbf{u}) = \mathbf{u}$$

and

$$Dg_{S'}(\mathbf{u}) = \mathbf{u}.$$

**Proof of claim.** Write any $\mathbf{u} \in \mathcal{U}$ uniquely as $\mathbf{u} = \sum_{j=1}^{k} a_j e_j$. Define $g_S(\mathbf{u}) \in \mathbb{R}^K$ by setting $(g_S(\mathbf{u}))_j = a_j$ for $j \in S$ and $(g_S(\mathbf{u}))_j = 0$ for $j \notin S$. Then $g_S(\mathbf{u})$ is supported on $S$ and $Dg_S(\mathbf{u}) = \sum_{j=1}^{k} a_j D_j = \sum_{j=1}^{k} a_j e_j = \mathbf{u}$. Continuity is immediate.

Next, define $g_{S'}(\mathbf{u}) \in \mathbb{R}^K$ as follows. For $j \in \{1, \ldots, k-2\}$ set $(g_{S'}(\mathbf{u}))_j = a_j$. For $j \notin S'$ set $(g_{S'}(\mathbf{u}))_j = 0$. It remains to set the two coordinates $(g_{S'}(\mathbf{u}))_{k+1}$ and $(g_{S'}(\mathbf{u}))_{k+2}$.

Let $\mathbf{b} = (a_{k-1}, a_k)^T \in \mathbb{R}^2$. Let $M \in \mathbb{R}^{2 \times 2}$ denote the matrix whose columns are the coordinates of $D_{k+1}$ and $D_{k+2}$ in the basis $(e_{k-1}, e_k)$. Then

$$M = \frac{1}{\sqrt{1+\varepsilon^2}} \begin{bmatrix} 1 & \varepsilon \\ \varepsilon & 1 \end{bmatrix}.$$

This matrix is invertible since its determinant equals $(1 - \varepsilon^2)/(1 + \varepsilon^2)$, which is nonzero for $\varepsilon \neq 1$. Define

$$\begin{bmatrix} (g_{S'}(\mathbf{u}))_{k+1} \\ (g_{S'}(\mathbf{u}))_{k+2} \end{bmatrix} = M^{-1} \mathbf{b}.$$

Then by construction we have

$$(g_{S'}(\mathbf{u}))_{k+1} D_{k+1} + (g_{S'}(\mathbf{u}))_{k+2} D_{k+2} = a_{k-1} e_{k-1} + a_k e_k.$$

Therefore $Dg_{S'}(\mathbf{u}) = \mathbf{u}$. Since $g_{S'}$ is linear in $\mathbf{u}$, it is continuous. This proves the claim.

Now return to the assumed encoder $f$. Define the reconstruction map

$$\mathbf{u}(\mathbf{x}) = Df(\mathbf{x}).$$

By the remark above, $\mathbf{u}(\mathbf{x}) \in \mathcal{U}$ for all $\mathbf{x} \in \mathcal{X}$. Define two candidate encoders on $\mathcal{X}$ by

$$f_S(\mathbf{x}) = g_S(\mathbf{u}(\mathbf{x}))$$

and

$$f_{S'}(\mathbf{x}) = g_{S'}(\mathbf{u}(\mathbf{x})).$$

Each map is continuous as a composition of continuous maps. Also, $f_S(\mathbf{x})$ is supported on $S$ and $f_{S'}(\mathbf{x})$ is supported on $S'$ for every $\mathbf{x} \in \mathcal{X}$, so both satisfy the $k$-sparsity constraint.

Further, for every $\mathbf{x} \in \mathcal{X}$ we have

$$Df_S(\mathbf{x}) = Dg_S(\mathbf{u}(\mathbf{x})) = \mathbf{u}(\mathbf{x}) = Df(\mathbf{x})$$

and similarly

$$Df_{S'}(\mathbf{x}) = Dg_{S'}(\mathbf{u}(\mathbf{x})) = \mathbf{u}(\mathbf{x}) = Df(\mathbf{x}).$$

Therefore both satisfy the same reconstruction error bound as $f$. For example,

$$\|Df_S(\mathbf{x}) - \mathbf{x}\| = \|Df(\mathbf{x}) - \mathbf{x}\| \leq \epsilon,$$

and the same holds for $f_{S'}$. Set $f'$ to be either $f_S$ or $f_{S'}$, which proves the first part of the lemma.

It remains to show the disagreement and different supports under the additional assumption. Suppose there exists $\mathbf{x}_0 \in \mathcal{X}$ such that $\mathbf{u}(\mathbf{x}_0)$ has a nonzero component along $\mathrm{span}\{e_{k-1}, e_k\}$. Writing $\mathbf{u}(\mathbf{x}_0) = \sum_{j=1}^{k} a_j e_j$, this means $(a_{k-1}, a_k) \neq (0, 0)$. Then $g_S(\mathbf{u}(\mathbf{x}_0))$ has a nonzero entry on at least one of the indices $k-1$ or $k$, while $g_{S'}(\mathbf{u}(\mathbf{x}_0))$ has a nonzero entry on at least one of the indices $k+1$ or $k+2$. In particular, $g_S(\mathbf{u}(\mathbf{x}_0)) \neq g_{S'}(\mathbf{u}(\mathbf{x}_0))$ and their supports are different.

If $f$ differs from $f_S$ at some point, choose $f' = f_S$ and set $\mathbf{x}_1$ to be any point where they differ. Otherwise $f$ equals $f_S$ everywhere, and then choosing $f' = f_{S'}$ and $\mathbf{x}_1 = \mathbf{x}_0$ gives $f(\mathbf{x}_1) \neq f'(\mathbf{x}_1)$ with different supports. This completes the proof. $\square$

Now, we prove our identifiability result. We enumerate all assumptions used in our construction.

**Assumptions**  Below, we enumerate all assumptions used in the lemmas and proofs. We assume both SAEs in the pair follow these assumptions.

(A1) **Approximate RIP.** For observed supports $S$ and $S'$, $\delta_{S \cup S'} \leq \delta$ for some $\delta < 1$. Intuitively, this means that RIP holds on the observed union of supports for any two inputs.

(A2) **Sufficient richness of supports.** For any $i \in [K]$, we observe supports $S$ and $S'$ such that $S \cap S' = \{i\}$. Furthermore, for any pair of concepts $i$ and $j$, there exist disjoint supports $S$ and $S'$ such that $i \in S$ and $j \in S'$. Intuitively, this means that no two concepts always co-occur.

(A3) **Bounded reconstruction error.** Observations can be decomposed as $\mathbf{x} = D\mathbf{z} + \boldsymbol{\epsilon} = D\mathbf{z}' + \boldsymbol{\epsilon}'$, and $\|\boldsymbol{\epsilon}\|, \|\boldsymbol{\epsilon}'\| \leq \epsilon < \frac{\zeta\sqrt{1-\delta}\,\alpha(\delta)}{4\sqrt{k}}$. Intuitively, this means the reconstruction error of each SAE is strictly bounded everywhere by a constant which depends on how well the aRIP constraint holds, how well the sufficient diversity constraint holds, and the sparsity level.

(A4) **Sufficient diversity of observations.** For any observed support $S$ of size $k$, we have $\mathrm{Cov}[\mathbf{Z}_S \mid S]$ is positive definite, and furthermore there exists a batch of $k$ samples from $\mathbf{Z}_S \mid S$ such that the $k \times k$ matrix of samples has smallest singular value larger than $\zeta$.

(A5) **Bounded data distribution.** The observation distribution $\mathbf{X}$ has bounded support, that is, there exists $B > 0$ such that $\|\mathbf{X}\| \leq B$ everywhere.

(A6) **Least-squares optimality.** For each observation $\mathbf{x} = D\mathbf{z} + \boldsymbol{\epsilon}$, we assume the codes $\mathbf{z}$ are least-squares optimal on the support set $S$.

Our first lemma shows that when $2k$-aRIP holds on any observed combination of supports, this forcibly separates the subspaces spanned by those supports. In the language of Figure 1, the span of any two distinct patches must be decomposable into its intersection and two approximately "unique" subspaces which are well-separated. On the other hand, if these two patches share no concepts, the patches themselves must be well-separated.

**Lemma A.1.** *Let $S$ and $S'$ be sparse supports of size $k$ such that $|S \cap S'| < k$. Suppose $\delta_{S \cup S'} \leq \delta < 1$. Then, there exists $\alpha(\delta) > 0$ such that $\sin \theta_{max}(\mathcal{U}_S, \mathcal{U}_{S'}) \geq \alpha(\delta)$ where $\mathcal{U}_S = span(D_S)$ and $\mathcal{U}_{S'} = span(D_{S'})$. Further, if $|S \cap S'| = 0$, we have $\sin \theta_{min}(\mathcal{U}_S, \mathcal{U}_{S'}) \geq \alpha(\delta)$.*

*Proof.* Denote the intersection by $I = S \cap S'$. Denote the isolated components of each support by $A = S \setminus I$ and $B = S' \setminus I$.

**Claim.** $\mathcal{U}_S \cap \mathcal{U}_{S'} = \mathcal{U}_I$.

**Proof of claim.** First, note that RIP implies that $D_S$, $D_{S'}$ and $D_{S \cup S'}$ all have full column rank. Let $\mathbf{x} \in \mathcal{U}_S \cap \mathcal{U}_{S'}$, noting that then there exist decompositions $\mathbf{x} = D_S \mathbf{z} = D_{S'} \mathbf{z}'$. As a result, we have:

$$\mathbf{0} = D_S \mathbf{z} - D_{S'} \mathbf{z}' = D_I(\mathbf{z}_I - \mathbf{z}'_I) + D_A \mathbf{z}_A - D_B \mathbf{z}_B = D_{S \cup S'} \mathbf{w}$$

where $\mathbf{w}$ stacks the individual components, and must be zero by the fact that $D_{S \cup S'}$ has full column rank. Thus $\mathbf{z}_I = \mathbf{z}'_I$ and $\mathbf{x} = D_I \mathbf{z}_I$, proving the claim.

Denote $\mathcal{U}_1 = \mathcal{U}_S \cap \mathcal{U}_I^\perp$ and $\mathcal{U}'_1 = \mathcal{U}_{S'} \cap \mathcal{U}_I^\perp$.

**Claim.** $\theta_{\max}(\mathcal{U}_S, \mathcal{U}_{S'}) \geq \theta_{\min}(\mathcal{U}_1, \mathcal{U}'_1)$.

**Proof of claim.** We have $\mathcal{U}_S = \mathcal{U}_1 \oplus \mathcal{U}_I^\perp$ and $\mathcal{U}_{S'} = \mathcal{U}'_1 \oplus \mathcal{U}_I^\perp$ with $\dim(\mathcal{U}_1) = \dim(\mathcal{U}'_1) = k - |I|$. This means the nonzero principal angles between $\mathcal{U}_S$ and $\mathcal{U}_{S'}$ are exactly the principal angles between $\mathcal{U}_1$ and $\mathcal{U}'_1$. In particular, we have the claim.

Denote by $P_I$ the orthogonal projector onto $\mathcal{U}_I$.

**Claim.** The projected dictionary $(I - P_I)D_{A \cup B}$ satisfies RIP at the same level $\delta$.

**Proof of claim.** Let $\mathbf{z}_{A \cup B} \in \mathbb{R}^{|A \cup B|}$ denote an arbitrary vector and let $\mathbf{z}_I$ denote the least-squares minimizer of $\|D_{A \cup B} \mathbf{z}_{A \cup B} - D_I \mathbf{z}_I\|_2$. Let $\mathbf{r} = D_{A \cup B} \mathbf{z}_{A \cup B} - D_I \mathbf{z}_I = (I - P_I)D_{A \cup B} \mathbf{z}_{A \cup B} \in \mathcal{U}_I^\perp$ be the residual. Stacking $\mathbf{z}_{A \cup B}$ and $-\mathbf{z}_I$ into $\mathbf{w} \in \mathbb{R}^{|S \cup S'|}$, we have $D_{S \cup S'} \mathbf{w} = D_{A \cup B} \mathbf{z}_{A \cup B} - D_I \mathbf{z}_I = \mathbf{r}$. Now, we have:

$$\|(I - P_I)D_{A \cup B}\mathbf{z}_{A \cup B}\|^2 = \|\mathbf{r}\|^2 = \|D_{S \cup S'}\mathbf{w}\|^2 \geq (1 - \delta)\|\mathbf{w}\|^2 \geq (1 - \delta)\|\mathbf{z}_{A \cup B}\|^2$$

and also by the fact that $\delta_{A \cup B} \leq \delta_{S \cup S'}$:

$$\|(I - P_I)D_{A \cup B}\mathbf{z}_{A \cup B}\|^2 \leq \|D_{A \cup B}\mathbf{z}_{A \cup B}\|^2 \leq (1 + \delta)\|\mathbf{z}_{A \cup B}\|^2$$

which yields the claim by the arbitrariness of $\mathbf{z}_{A \cup B}$.

**Claim.** $\sin \theta_{\min}(\mathcal{U}_1, \mathcal{U}_1') \geq \alpha(\delta)$ for $\alpha(\delta) = \sqrt{1 - \left(\frac{2\delta}{1+\delta}\right)^2}$.

**Proof of claim.** Let $\mathbf{u}_S$ and $\mathbf{u}_{S'}$ be unit vectors realizing the minimum principal angle $\langle \mathbf{u}_S, \mathbf{u}_{S'} \rangle = \cos \theta_{\min}(\mathcal{U}_S, \mathcal{U}_{S'})$. Then $\mathbf{u}_S = (I - P_I)D_S\mathbf{z}_S$ and $\mathbf{u}_{S'} = (I - P_I)D_{S'}\mathbf{z}_{S'}$. Then,

$$\|\mathbf{u}_S - \mathbf{u}_{S'}\|^2 \geq (1 - \delta)(\|\mathbf{z}_S\|^2 + \|\mathbf{z}_{S'}\|^2) \geq 2(1 - \delta)/(1 + \delta)$$

where the first inequality follows by hypothesis and the second from the previous claim. Rearranging the usual definition of cosine similarity for unit vectors, we have $\cos \theta_{\min}(\mathcal{U}_S, \mathcal{U}_{S'}) \leq 2\delta/(1 + \delta)$. By the Pythagorean identity, we have the claim.

Using the second claim, the first statement of the lemma follows. If $S \cap S' = \emptyset$, then $\mathcal{U}_1 = \mathcal{U}_S$ and $\mathcal{U}_1' = \mathcal{U}_{S'}$ so the latter statement of the lemma follows. $\qquad \square$

Our second lemma shows that a given support in the first SAE must in some sense map uniquely to a single support in the second SAE, where uniqueness is defined in the same way as in the separation argument in the previous lemma. In the language of Figure 1, this lemma shows that if patches are well-separated in a pair of SAEs, low reconstruction error is enough to "adhere" a particular linear patch to a region of the observation manifold, and this linear patch can't look too different from the linear patch approximating that region of the manifold in the second SAE.

**Lemma A.2.** *Let $\mathbf{x}^{(i)} = D\mathbf{z}^{(i)} + \boldsymbol{\epsilon}^{(i)}$ denote $k$ distinct $k$-sparse decompositions with sparse support $S$ under the first SAE satisfying (A4), with smallest singular value bounded below by $\zeta$. Further, assume that these observations also admit a decomposition under the second SAE, $\mathbf{x}^{(i)} = D'\mathbf{z}'^{(i)} + \boldsymbol{\epsilon}'^{(i)}$, with sparse support $T^{(i)}$ with $|T^{(i)}| = k$. Let $\mathcal{T}_S$ be the set of supports that occur in the second SAE, and suppose*

$$\sin \theta_{max}(\mathcal{V}_T, \mathcal{V}_{T'}) \geq \beta > 0$$

*for all distinct $T, T' \in \mathcal{T}$. If $\epsilon < \frac{\sqrt{1-\delta}\zeta\beta}{4\sqrt{k}}$, then all supports coincide: $T^{(1)} = \cdots = T^{(k)} = T$ for some $T$. Furthermore,*

$$\sin \theta_{max}(\mathcal{U}_S, \mathcal{V}_T) \leq \frac{2\sqrt{k}}{\sqrt{1-\delta}\zeta}\epsilon$$

*and for every $T' \neq T$ in $\mathcal{T}_S$,*

$$\sin \theta_{max}(\mathcal{U}_S, \mathcal{V}_{T'}) \geq \beta/2$$

*Proof.* Denote the reconstructions under the first SAE as $\hat{\mathbf{x}}^{(i)} = D_S\mathbf{z}_S^{(i)}$. Our first claim is that if all $T^{(i)}$ are actually a single support $T$, we have that $\mathcal{U}_S$ is near $\mathcal{V}_T$.

**Claim.** Suppose $T^{(1)} = \cdots = T^{(k)} = T$ for some support $T$. Then

$$\sin \theta_{\max}(\mathcal{U}_S, \mathcal{V}_T) \leq \frac{2\sqrt{k}}{\zeta\sqrt{1-\delta}}\epsilon$$

**Proof of claim.** Applying the triangle inequality, we have the tube constraint that the reconstruction under the first SAE cannot be far from $\mathcal{V}_T$: $\|(I - P_T)\hat{\mathbf{x}}^{(i)}\| \leq 2\epsilon$. Stacking the reconstructions into a matrix $\hat{X}$, we have $\|(I - P_T)\hat{X}\|_{op} \leq$

$\|(I - P_T)\hat{X}\|_F \le 2\epsilon\sqrt{k}$. Let $Q$ be an orthonormal basis matrix for $\mathcal{U}_S$. Write $\hat{X} = QR$, where $R$ is invertible by the fact that $D_S$ is full rank and (A4), and furthermore $\sigma_{\min}(\hat{X}) \ge \sigma_{\min}(D_S Z) \ge \sigma_{\min}(D_S)\sigma_{\min}(Z) \ge \zeta\sqrt{1 - \delta}$, thus

$$\sin\theta_{\max}(\mathcal{U}_S, \mathcal{V}_T) = \|(I - P_T)Q\|_{\text{op}} \le \|(I - P_T)\hat{X}\|_{\text{op}}\|R^{-1}\|_{\text{op}} \le \frac{2\epsilon\sqrt{k}}{\zeta\sqrt{1 - \delta}}$$

where the denominator in the final inequality follows from the fact that $\sigma_{\min}(\hat{X}) = 1/\sigma_{\min}(R^{-1})$.

Next, we consider the case where the supports in the second SAE do not coincide.

**Claim.** $T^{(1)} = \cdots = T^{(k)} = T$ for some support $T$.

**Proof of claim.** Consider toward a contradiction that $T^{(i)} \ne T^{(j)}$ for some $i$ and $j$. Noting that the argument in the previous claim only hinges on the projection operator $P_{T^{(i)}}$, we can apply the same argument to $T^{(i)}$ and $T^{(j)}$ independently, combining the results to obtain

$$\sin\theta_{\max}(\mathcal{V}_{T^{(i)}}, \mathcal{V}_{T^{(j)}}) \le \sin\theta_{\max}(\mathcal{U}_S, \mathcal{V}_{T^{(i)}}) + \sin\theta_{\max}(\mathcal{U}_S, \mathcal{V}_{T^{(j)}}) \le \frac{4\epsilon\sqrt{k}}{\zeta\sqrt{1 - \delta}}$$

which contradicts the hypotheses $\sin\theta_{\max}(\mathcal{V}_{T^{(i)}}, \mathcal{V}_{T^{(j)}}) \ge \beta$ and $\epsilon < \frac{\sqrt{1-\delta}\zeta\beta}{4\sqrt{k}}$.

Together, these claims give the main conclusion. The second conclusion follows from the fact that for any other $T'$, we have

$$\sin\theta_{\max}(\mathcal{U}_S, \mathcal{V}_{T'}) \ge \sin\theta_{\max}(\mathcal{U}_S, \mathcal{V}_T) - \sin\theta_{\max}(\mathcal{V}_T, \mathcal{V}_{T'}) \ge \beta - \frac{2\epsilon\sqrt{k}}{\zeta\sqrt{1 - \delta}} \ge \beta/2$$

where the final bound follows from the hypothesis on $\epsilon$. $\qquad\square$

With these two lemmas in hand, we can prove our identifiability theorem.

*Theorem* (3.6, formal). Let $\mathbf{X}$ denote a random observation, and consider a pair of trained sparse autoencoders $(f, D)$ and $(f', D')$. Denote the sparse codes $\mathbf{Z} = f(\mathbf{X})$ and $\mathbf{Z}' = f'(\mathbf{X})$, and suppose we have $\mathbf{X} = D\mathbf{Z} + \boldsymbol{\epsilon} = D'\mathbf{Z}' + \boldsymbol{\epsilon}'$ where $\|\boldsymbol{\epsilon}\|, \|\boldsymbol{\epsilon}'\| \le \epsilon$ for $\epsilon$. Furthermore, suppose that the approximate RIP assumption (A1) is satisfied for both models, that the observed support distributions in both models are sufficiently rich to isolate concepts (A2), that the reconstruction error in both models is bounded (A3), and the observation distribution is bounded (A5) sufficiently diverse to witness all dimensions of each latent support (A4). Finally, assume that the models are sufficiently trained such that the sparse codes satisfy the least-squares solution on their supports (A6). Then, there exists a signed permutation $\pi$ (or $\Pi$, in matrix form) such that we have:

1. **Dictionary near-identifiability.** $\|\mathbf{d}_i - \mathbf{d}'_{\pi(i)}\| \le 2C_1(\delta)\eta$ for $C_1(\delta) = 1 + 2/\alpha(\delta)$

2. **Code near-identifiability.** $\|\mathbf{Z} - \Pi\mathbf{Z}'\| \le \frac{2\epsilon}{\sqrt{1-\delta}} + \frac{(2\sqrt{k}C_1(\delta)\eta)(B+\epsilon)}{1-\delta}$

where $\eta = \frac{2\sqrt{k}}{\zeta\sqrt{1-\delta}}\epsilon$ and $\alpha(\delta) = \sqrt{1 - \left(\frac{2\delta}{1+\delta}\right)^2}$.

*Proof.* We begin by showing that for a given support in the first SAE, there is only one sparse support in second SAE that can accurately represent the observations from that support.

**Claim.** For a sparse support $S$ in the first SAE, we have a well-defined support map $T = \Phi(S)$ where $T$ is a sparse support in the second SAE. Furthermore, we have $\sin\theta_{\max}(\mathcal{U}_S, \mathcal{V}_T) \le \frac{2\sqrt{k}}{\zeta\sqrt{1-\delta}\epsilon} =: \eta$.

**Proof of claim.** Using (A4), select a batch of $k$ observations $\mathbf{x}^{(i)}$ with the support $S$ such that their coefficient matrix $Z$ satisfies $\sigma_{\min}(Z) \ge \zeta$. Note that each observation also admits a $k$-sparse decomposition with appropriately bounded error in the second SAE, supported on $T^{(i)}$. By Lemma A.1, we have that for $T \ne T' \in \mathcal{T}_S$, the set of these supports, $\sin\theta_{\max}(\mathcal{V}_T, \mathcal{V}_{T'}) \ge \sqrt{1 - \left(\frac{2\delta}{1+\delta}\right)^2} =: \alpha(\delta)$. By the fact that $\epsilon < \frac{\zeta\sqrt{1-\delta}\alpha(\delta)}{4\sqrt{k}}$, Lemma A.2 implies that all the supports coincide: $T := T^{(1)} = \cdots T^{(k)}$, along with the bound. From here on, we set $T = \Phi(S)$.

Next, we use this support-by-support mapping to show the existence of a concept-by-concept mapping between the two SAEs. Pick a concept $i$ in the first SAE. By the sufficient richness assumption (A2), we can select two supports $S_1$ and $S_2$ such that $S_1 \cap S_2 = \{i\}$. Let $T_1 = \Phi(S_1)$ and $T_2 = \Phi(S_2)$ be the corresponding supports in the second SAE as given in the previous claim, satisfying $\sin \theta_{\max}(\mathcal{U}_{S_\ell}, \mathcal{V}_{T_\ell}) \leq \eta$.

Now, we need a technical claim which bounds the distance between the two supports in the same SAE in terms of the distance from the intersection.

**Claim.** For every $\hat{\mathbf{x}} \in \mathcal{U}_{S_1}$, we have $\|(I - P_{S_2})\hat{\mathbf{x}}\| \geq \alpha \|(I - P_I)\hat{\mathbf{x}}\|$.

**Proof of claim.** Denote $\mathcal{U}_I = \mathrm{span}(\mathbf{d}_i)$, and define the residual spaces $\mathcal{U}_1 = \mathcal{U}_{S_1} \cap \mathcal{U}_I^\perp$ and $\mathcal{U}_2 = \mathcal{U}_{S_2} \cap \mathcal{U}_I^\perp$. By the second claim in Lemma A.1, we have $\sin \theta_{\min}(\mathcal{U}_1, \mathcal{U}_2) \geq \alpha(\delta)$. For any $\hat{\mathbf{x}} \in \mathcal{U}_{S_1}$,

$$\|(I - P_{S_2})\hat{\mathbf{x}}\| = \|(I - P_{\mathcal{U}_2})(I - P_I)\hat{\mathbf{x}}\| \geq \alpha(\delta)\|(I - P_I)\hat{\mathbf{x}}\|$$

Now, we study the other SAE. Our first goal is to show that like in the first SAE, these supports must share at least one concept.

**Claim.** $T_1 \cap T_2 \neq \emptyset$

**Proof of claim.** Assume toward a contradiction that $T_1 \cap T_2 = \emptyset$. Note that Lemma A.1 yields $\sin \theta_{\min}(\mathcal{V}_{T_1}, \mathcal{V}_{T_2}) \geq \alpha(\delta)$. Pick $\mathbf{v}$ in $\mathcal{V}_{T_1}$ such that it's closest to $\mathbf{d}_i$, then by the triangle inequality we have

$$\|(I - P_{T_2})\mathbf{v}\| \leq \|\mathbf{v} - \mathbf{d}_i\| + \|(I - P_{T_2})\mathbf{d}_i\| \leq 2\eta$$

which contradicts the minimum principal angle bound.

Now, we show that the corresponding supports in the other SAE share exactly one concept.

**Claim.** $|T_1 \cap T_2| = 1$

**Proof of claim.** Assume toward a contradiction that $J = T_1 \cap T_2$ satisfies $|J| \geq 2$. Now, given $\mathcal{V}_J = \mathcal{V}_{T_1} \cap \mathcal{V}_{T_2}$, pick a unit vector $\mathbf{v}$ in $\mathcal{V}_J$ such that $\mathbf{v} \perp \mathbf{d}_i$. By the previous claim, we can pick $\mathbf{u}_1 \in \mathcal{U}_{S_1}$ $\eta$-close to $\mathbf{v}$ and $\mathbf{u}_2 \in \mathcal{U}_{S_2}$, so $\|\mathbf{u}_1 - \mathbf{u}_2\| \leq \|\mathbf{u}_1 - \mathbf{v}\| + \|\mathbf{u}_2 - \mathbf{v}\| \leq 2\eta$ by the triangle inequality.

Now, we have $\|(I - P_I)\mathbf{v}\| = 1$ by the perpendicularity, and therefore $\|(I - P_I)\mathbf{u}_1\| \geq 1 - \eta$. Thus, by our previous technical claim we have $\|(I - P_{S_2})\mathbf{u}_1\| \geq \alpha(\delta)(1 - \eta)$. Combining, we have $2\eta \geq \|\mathbf{u}_1 - \mathbf{u}_2\| \geq \|(I - P_{S_2})\mathbf{u}_1\| \geq \alpha(\delta)(1 - \eta)$ which contradicts the $\epsilon$ bound. Therefore $|J| = 1$.

Because in the first SAE $S_1$ and $S_2$ share a single concept, and $T_1$ and $T_2$ also share a single concept, we posit a mapping between the two. However, we have to show that this mapping is well-defined, in the sense that a different pair of supports in the first SAE sharing the same concept maps to the same concept in the second SAE.

**Claim.** Defining $\pi(i)$ as the unique element of $T_1 \cap T_2$ gives a well-defined mapping $\pi : [K] \to [K]$, independent of the particular choices of $S_1$ and $S_2$.

**Proof of claim.** Take any alternative choice $\tilde{S}_1$ and $\tilde{S}_2$, and apply the previous two claims to obtain $\tilde{T}_1 = \Phi(\tilde{S}_1)$ and $\tilde{T}_2 = \Phi(\tilde{S}_2)$ with $\tilde{T}_1 \cap \tilde{T}_2 = \{\tilde{j}\}$. Take $\mathbf{v} \in \mathcal{V}_{T_1}$ such that $\|\mathbf{v} - \mathbf{d}_i\| \leq \eta$, by Lemma A.2 applied to $\mathcal{U}_{S_1}$ and $\mathcal{V}_{T_1}$. Then, we have

$$\|(I - P_{T_2})\mathbf{v}\| \leq \|(I - P_{T_2})\mathbf{d}_i\| + \|\mathbf{v} - \mathbf{d}_i\| \leq 2\eta$$

yielding $\|(I - P_j)\mathbf{v}\| \leq 2\eta/\alpha(\delta)$ via Lemma A.1. Thus, we have

$$\|(I - P_j)\mathbf{d}_i\| \leq \|\mathbf{d}_i - \mathbf{v}\| + \|(I - P_j)\mathbf{v}\| \leq (1 + 2/\alpha(\delta))\eta$$

and a similar argument yields $\|(I - P_{\tilde{j}})\mathbf{d}_i\| \leq (1 + 2/\alpha(\delta))\eta := \rho$. Suppose now toward a contradiction that $j \neq \tilde{j}$. Choosing signs $s$ and $s'$ appropriately, we have $\|\mathbf{d}_i - s\mathbf{d}'_j\| \leq 2\rho$ and $\|\mathbf{d}_i - s\mathbf{d}'_{\tilde{j}}\| \leq 2\rho$. Applying the triangle inequality gives $\langle \mathbf{d}'_j, \mathbf{d}'_{\tilde{j}} \rangle \geq 1 - 8\rho^2$, i.e. $\rho \geq \sqrt{(1 - \delta)/8}$ by restricting the RIP condition (A1). This yields a contradiction given our $\epsilon$ bound.

As a result, we can define $\pi(i) = j$. It remains to show that it's a permutation, which covers the same edge case as in the previous claim but in reverse.

**Claim.** $\pi$ is a permutation of $[K]$.

**Proof of claim.** Suppose toward a contradiction $\pi$ is not injective, with $\pi(i) = \pi(i') = j$ with $i \neq i'$ for $i \in S$ and $i' \in S'$. For $T = \Phi(S)$ and $T' = \Phi(S')$, we have $\sin\theta_{\min}(\mathcal{V}_T, \mathcal{V}_{T'}) = 0$ by the fact that $\mathrm{span}(\mathbf{d}'_j) \subset \mathcal{V}_T \cap \mathcal{V}_{T'}$. On the other hand, Lemma A.1 gives $\sin\theta_{\min}(\mathcal{U}_S, \mathcal{U}_{S'}) \geq \alpha(\delta)$. By the triangle inequality and the fact that $\sin\theta_{\max}(\mathcal{U}_S, \mathcal{V}_T) \leq \eta$ and $\sin\theta_{\max}(\mathcal{U}_{S'}, \mathcal{V}_{T'}) \leq \eta$, we have $\sin\theta_{\min}(\mathcal{V}_T, \mathcal{V}_{T'}) \geq \alpha(\delta) - 2\eta$. Given our $\epsilon$ bound, this is a contradiction. Thus $\pi$ is injective, and given that the domain and codomain are finite and of the same size, a permutation.

The hard part is done. Now, using the mapping $\pi$ we constructed, we can show individual concepts in the dictionary are near-identifiable up to signs, according to the permutation $\pi$.

**Claim.** For either $s_i = +1$ or $s_i = -1$, we have $\|\mathbf{d}_i - s_i\mathbf{d}'_{\pi(i)}\| \leq 2C_1(\delta)\eta$ for $C_1(\delta) = 1 + 2/\alpha(\delta)$.

**Proof of claim.** Fix $i$ and pick $S_1$ and $S_2$ with $S_1 \cap S_2 = \{i\}$. Set $T_1 = \Phi(S_1)$ and $T_2 = \Phi(S_2)$, by the previous claims we have $T_1 \cap T_2 = \{\pi(i)\}$. For any $\mathbf{v} \in \mathcal{V}_{T_1}$, we have

$$\|(I - P_{T_2})\mathbf{v}\| \geq \alpha(\delta)\|(I - P_{\pi(i)})\mathbf{v}\|$$

where $P_{\pi(i)}$ is the projection matrix onto $\mathcal{V}_{\pi(i)} = \mathrm{span}(\mathbf{d}'_{\pi(i)})$. Select $\mathbf{w} \in \mathcal{V}_{T_1}$ such that $\|\mathbf{d}_i - \mathbf{w}\| \leq \eta$. By the triangle inequality, we have $\|(I - P_{T_2})\mathbf{v}\| \leq \|(I - P_{T_2})\mathbf{d}_i\| + \|\mathbf{d}_i - \mathbf{v}\| \leq 2\eta$. As a result, we have $\|(I - P_{\pi(i)})\mathbf{v}\| \leq 2\eta/\alpha(\delta)$. Thus,

$$\|(I - P_{\pi(i)})\mathbf{d}_i\| \leq \|\mathbf{d}_i - \mathbf{v}\| + \|(I - P_{\pi(i)})\mathbf{v}\| \leq (1 + 2/\alpha)\eta$$

with the final bound following by the definition of point-set distance in terms of sines.

Finally, support stability and dictionary stability yields code stability.

**Claim.** For any observation $\mathbf{x} = D\mathbf{z} + \boldsymbol{\epsilon} = D'\mathbf{z}' + \boldsymbol{\epsilon}'$, we have $\|\mathbf{z} - \Pi\mathbf{z}'\| \leq \|\mathbf{z} - \Pi\mathbf{z}'\| \leq \frac{2\epsilon}{\sqrt{1-\delta}} + \frac{(2\sqrt{k}\,C_1(\delta)\,\eta)(B+\epsilon)}{1-\delta} = \mathcal{O}(\epsilon)$.

**Proof of claim.** Let $S$ denote the support of $\mathbf{z}$, and $T$ denote the support $\mathbf{z}'$. We have,

$$\|\mathbf{z} - \Pi\mathbf{z}'\| = \|\mathbf{z}_S - \Pi_T\mathbf{z}'_T\|$$
$$\leq \|\mathbf{z}_S - D_S^\dagger\hat{\mathbf{x}}_T\| + \|D_S^\dagger\hat{\mathbf{x}}_T - \Pi_T\mathbf{z}'_T\|$$

where $D_S^\dagger$ is the left inverse of $D_S$, well-conditioned by (A1). Denote by $\bar{D}_S = D'\Pi_S$ be the sign- and permutation-matched dictionary from the second SAE. Then, we have $D_S^\dagger\hat{\mathbf{x}}_T - \Pi_T\mathbf{z}'_T = D_S^\dagger(\bar{D}_S - D_S)(\Pi_S\mathbf{z}'_T)$. As a result,

$$\|\mathbf{z} - \Pi\mathbf{z}'\| \leq \|\mathbf{z}_S - D_S^\dagger\hat{\mathbf{x}}_T\| + \|D_S^\dagger\hat{\mathbf{x}}_T - \Pi_T\mathbf{z}'_T\|$$
$$\leq \frac{2\epsilon}{\sqrt{1-\delta}} + \|D_S^\dagger\|_{\mathrm{op}}\|\bar{D}_S - D_S\|_{\mathrm{op}}\|\Pi_S\mathbf{z}'_T\|$$
$$\leq \frac{2\epsilon}{\sqrt{1-\delta}} + \frac{1}{\sqrt{1-\delta}}(2\sqrt{k}C_1(\delta)\eta)\frac{B+\epsilon}{\sqrt{1-\delta}}$$
$$= \frac{2\epsilon}{\sqrt{1-\delta}} + \frac{(2\sqrt{k}C_1(\delta)\eta)(B+\epsilon)}{1-\delta}$$

which is $\mathcal{O}(\epsilon)$, the same rate we would hope for if we knew the dictionaries and supports perfectly in advance. Furthermore, the remainder of the constants could be made tighter by more careful treatment of the second term. $\square$

