# OpenReview forum: "Toward Identifiable Sparse Autoencoders"
_ICML.cc/2026/Conference — ICML 2026 regular_

### Official Review · Reviewer_JzEy · 2026-03-07

**Soundness:** 3
**Presentation:** 2
**Significance:** 2
**Originality:** 2
**Overall Recommendation:** 4
**Confidence:** 3

**Summary:**

This paper elaborates on sparse auto-encoders, an encoder-decoder neural architecture, where the latent representation is sparse. To address limitations of SAEs like their instability in finding the desired solution, the authors propose to couple SAEs with dictionary learning, since the two frameworks share common characteristics like the underlying sparsity.

To that end, the authors directly alter the architecture of a baseline SAE: first, they construct a dictionary for the SAE, and prove that it can yield non-identifiable solutions; secondly, they prove that when the chosen baseline is combined with a learned dictionary satisfying an approximate restricted isometry property - proposed by the authors - the SAE yields nearly-identifiable solutions. The overall proposed model, called iSAE, outperforms the baseline, as this is depicted by relevant experiments.

**Compliance With Llm Reviewing Policy:**

Affirmed.

**Final Justification:**

The authors have addressed my comments, so I'll raise my score.

**Key Questions For Authors:**

1. Could you provide more details of related work, what's missing, what's at stake, and how your work contributes to that direction? All these should be addressed in the first 1-1.5 pages, to directly allow the reader to catch up with your proposed method.
2. Some polishing is required here and there: for instance, it's not elegant to add a mathematical theorem in the main results section and state it as "Informal" (see Theorem 3.2). This is the time and place where you state your work in a formal way, if not here, where else should a formal theorem be stated??? This also happens with different names given in the proofs in the appendix and the main paper, where the reader has to guess that Theorem 3.2 actually pertains to Theorem A.4. Moreover, Theorem 3.1 looks quite narrative. Perhaps boost it with some more mathematical notation?
3. Could you please give more references, or clarifications, w.r.t. several notions or claims? These include:
     a) in Sec. 2, p.1, you mention the TopK SAEs for the first time. What are these?
     b) in p. 2, right after (2), this paragraph is not properly justified. These "traditional approaches" solve a relaxation of (2), under an $\ell_1$ norm penalty, due to the NP-hardness of the problem, induced by the $\ell_0$ norm. It's not just the conditions on D that render (2) unsolvable. Please amend this part accordingly.
     c) in the first paragraph of Sec. 3.1: the claim you make on the first half is not adequately supported. Could you added more references?

**Limitations:**

Yes.

**Strengths And Weaknesses:**

## Strengths

The paper is well-motivated, with relevant claims. It features theoretical results, which are supported by relevant experiments. Importantly, the authors provide a thorough discussion around their experimental findings, highlighting the potential of the results.

## Weaknesses

1. The paper feels a bit rushed: the first page quickly introduces everything, without further information to what's at stake. Then, relevant information are given here and there, but not in the right proportion, to allow the reader to actually read the paper sequentially. ICML papers are supposed to be 8 pages long, so there's plenty of space that's not taken advantage of.
2. The originality of the proposed framework is mildly convincing. For instance, the proposed aRIP reminds of standard variants of the RIP (also, there's a typo in the RIP: the squares should either appear in all sides of the inequalities, or none). Not enough explanations and clarifications are put to bolster the paper's significance. For instance, the proposed method is called iSAE, but there's not a formal definition throughout the text pertaining to what iSAE precisely is.
3. The overall presentation makes the paper difficult to follow (related to my first point above).

---

> ### Author Rebuttal · Authors · 2026-03-31
>
> Thank you for your kind words and attention to detail.
>
> **(W2) aRIP novelty**
>
> We briefly comment on the novelty of our theory. We think about our aRIP condition as a relaxation of the RIP condition, as you correctly point out. The key novelty is that our approximate RIP condition (with respect to a particular data distribution) allows for identifiability of models trained on that data distribution. To our knowledge, this is a completely novel application of the RIP theory to identifiability in a statistical setting, but we would welcome references to similar or relevant work in dictionary learning or sparse coding if you have them.
>
> **(KQ1) Additional related work**
>
> We have updated the Related Work section:
>
> > A growing body of recent work applies SAEs as a tool for post-hoc interpretability, including mechanistic interpretability, particularly in large language models (Elhage et al., 2022; Olah et al., 2020; Bricken et al., 2023). In these settings, SAEs are trained on intermediate activations (e.g., residual streams or MLP layers), and individual dictionary elements are interpreted as corresponding to human-meaningful concepts such as semantic topics, syntactic patterns, or behavioral circuits. This approach has enabled analyses of feature superposition (Elhage et al., 2022), circuit structure (Olah et al., 2020), and the localization of model behaviors, as well as interventions in which specific features are ablated or amplified to steer model outputs (Bricken et al., 2023; Turner et al., 2024). These applications rely on the empirical observation that SAE features are often sparse, localized, and partially interpretable, even in highly overcomplete regimes, provided that reconstruction performance is good enough.
> >
> > ...
> >
> > In most works on mechanistic interpretability, identifiability is assessed subjectively, by checking the alignment with human-interpretable concepts (Karvonen et al., 2025). In contrast, the historical view in sparse coding and dictionary learning is to assess when signals admit a stable sparse linear decomposition, without assigning a particular interpretation to it. Our work aims to bridge this gap by explicitly
> analyzing the conditions under which the representations learned by SAEs are stable and statistically identifiable across independent training runs, and by proposing metrics that capture this stability at the level
> of both dictionaries and codes.
>
> and added additional context about prior work applying SAEs.
>
> **(KQ2) Polishing**
>
> Thank you for your comments on polish. We have fixed a number of typos in the manuscript, as detailed in the response to *Reviewer VvMk*.
>
> **(KQ3) Clarifications**
>
> *(a) TopK definition*
>
> We have been clearer in the “Sparse autoencoding” section (following equation (1)) that TopK SAEs are a particular instance of equation (1) where $\sigma$ is the TopK function.
>
> *(b) $\ell_1$/$\ell_0$ distinction*
>
> Under our definition of identifiability (which you correctly note implies the ability to cast the problem as a well-posed optimization problem over a single loss/Lagrangian formulation; not true for the $\ell_0$ formulation), we have revised our identifiability statement:
>
> > Optimization is generally intractable due to the highly non-convex nature of the $\ell_0$ norm and the corresponding NP-hardness of the problem. Furthermore, even when it can be relaxed to the $\ell_1$ form, without conditions on $D$, the solution to the relaxation of (2) is known to be non-identifiable for general $x$.
>
> And:
>
> > Most algorithms, such as K-SVD (Aharon et al., 2006), alternately solve an $\ell_1$ relaxation of (2) for numerous samples and update the dictionary.
>
> *(c) Additional references in first half of first para of 3.1*
>
> We have added the following references to support the following claims:
>
> > Sparse autoencoders have several appealing properties. They are easily implemented in modern deep learning frameworks and trained on modern hardware using stochastic gradient descent (Gao et al., 2024; Karvonen et al., 2025; Fel et al., 2025). For large language models in particular, their lightweight architecture means that a large cache of input activations can be inferred while the SAE is trained “online”, allowing training to scale to billions of tokens for the largest models (Karvonen et al., 2025; Gao et al., 2024).
>
> Additional references not already in the paper:
>
> 1. Karvonen et al. SAEbench: A comprehensive benchmark for sparse autoencoders in language model interpretability, *ICML 2025*.
> 2. Elhage et al. Toy models of superposition, *Transformer Circuits Thread, 2022*.
> 3. Olah et al. Zoom in: An introduction to circuits. *Distill, 2020*.
> 4. Turner et al. Steering Language Models with Activation Engineering. 2023. https://arxiv.org/abs/2308.10248
> 5. Bricken et al. Towards monosemanticity: Decomposing language models with dictionary learning. *Transformer Circuits Thread, 2023*.

---

> > ### Author Rebuttal · Reviewer_JzEy · 2026-04-03
> >
> > Thank you for your rebuttal, I'll raise my score. Good luck with your submission!

---

### Official Review · Reviewer_Vw1r · 2026-03-12

**Soundness:** 3
**Presentation:** 2
**Significance:** 3
**Originality:** 3
**Overall Recommendation:** 4
**Confidence:** 3

**Summary:**

The paper addresses stability problems of sparse autoencoders (SAE's) and brings tools from dictionary learning and sparse coding to mitigate them. The authors follow the restricted isometry property (RIP) of Candes et al 2005 which ensures sparse coders to be identifiable. They attempt to approximately achieve the RIP property and introduce an approximated measure that is then used in finding a more stbale dictionary, they refer to as idenfifiable SAE. The main theoretical contribution is Th. 3.3 where thay show that under certain assumption the proposed aRIP regularization yields in different trainings almost identical dictionaries, upto a permutation matrix, where as regularization is stricter the dictionaries converge. The work follows by some synthetic and real examples, showing that using the proposed regularizer improves the SAE.

**Compliance With Llm Reviewing Policy:**

Affirmed.

**Final Justification:**

The authors answered parts of my concerns. However, regarding presentation and clarity of the appendix, I do not have the new version to verify this is indeed improved. Therefore maintaining my score.

**Key Questions For Authors:**

1. Near eq. (1), what is H? dimensions of b and x seem not to agree.
2. Above Th. 1, is it a theorem or a lemma?
3. Where is the proof of Th. 3.3, is it Th. A.4? the appendix in general is not well presented. The rationale and logic of the proofs are not given. A lot of claims just appear without any explanation why they are important and how to connect everything.

**Limitations:**

The authors have discussed the limitations adaquately.

**Strengths And Weaknesses:**

Strengths: logical connection between SAE's and dictionary learning. The paper improves the understanding of SAE's from both a theoretical and applied perspective.
Weaknesses: Presentation. Not everything is that clearly written, proofs in the appendix are quite messy and hard to follow. Experiments may be more comprehensive with additional data and methods.

---

> ### Author Rebuttal · Authors · 2026-03-31
>
> Thank you for your kind words and attention to detail.
>
> As discussed in our response to *Reviewer VvMk*, we have taken your concerns about presentation and clarity seriously. We have addressed the typos mentioned by both of you.
>
> **(KQ1) Notation: $H$**
>
> $H = N$, this is a typo that has been corrected. This pre-/post-bias formulation was standardized in the “Scaling TopK SAEs” paper (Gao et al. 2024), although we find our results are not particularly sensitive to the bias formulation we use. We will add a note clarifying this as well.
>
> **(KQ2) Impossibility result**
>
> This is best regarded as a theorem in our opinion, as it’s not used for the proof of the subsequent identifiability result. The errant reference in the text has been corrected.
>
> **(KQ3) Proof strategy**
>
> Thank you for the specific comments on our proof. We have now correctly labeled the theorems and proofs in the appendix, and before each claim we explain intuitively the role of that claim in the proof. We have also added further intuition about the two lemmas and the proof of the main theorem before each.

---

> > ### Author Rebuttal · Reviewer_Vw1r · 2026-04-04
> >
> > The authors answered parts of my concerns. However, regarding presentation and clarity of the appendix, I do not have the new version to verify this is indeed improved. Therefore maintaining my score.

---

### Official Review · Reviewer_m835 · 2026-03-12

**Soundness:** 3
**Presentation:** 2
**Significance:** 3
**Originality:** 3
**Overall Recommendation:** 4
**Confidence:** 3

**Summary:**

This paper provides theoretical grounding for identifiability limitations of SAEs and propose an improved variant (iSAE) with bidirectional features, multistep encoder architecture, and novel dictionary conditioning. The authors also provide principled theoretical backing for dictionary conditioning and empirical validation.

**Compliance With Llm Reviewing Policy:**

Affirmed.

**Final Justification:**

The paper presents clear theoretical grounding of a known significant empirical limitation for SAEs, identifiability, and provides an original and principled solutions backed by sound empirical evaluation. The paper and its results are well presented and in general easy to follow, the authors have fully addressed concerns regarding presentation in the rebuttal. In addition, most of the questions and weaknesses around empirical evaluation have also been addressed in the rebuttal.

**Key Questions For Authors:**

**Questions**
- What is the distribution of samples across concepts?
- How many SAEs for each variant?
- The permutations in Theorem 3.3 is not clearly explained.

**Limitations:**

- Limitations on empirical eval needs to be stated.

**Strengths And Weaknesses:**

**Strengths**
- the paper is well written and generally easy to follow,
- the paper presents clear theoretical grounding of a known significant empirical limitation for SAEs, identifiability, and provides principled solutions

**Weaknesses**
- **Definition of identifiability**: as the central problem tackled by this paper, identifiability should be defined first and in the main body
- **Clarity of assumptions**: the key assumptions for the principled dictionary conditioning should be stated in the main body
- **Limited empirical evaluation**: the comparative SAEs for empirical results are limited to topK and AbsTopK; the data is limited to synthetic iid, mixture data and pythia activations. The former may not be representative of variants of SAEs existing today and the latter is a 160M variant that may not be representative of standard LLM activations.
- **Ablations**: missing an ablation combination of AbsTopK + multistep encoding to empirically validate the principled dictionary conditioning.
- **Evaluation limited to metrics**: the purpose of the SAEs are for mechanistic interpretability and control, metrics of MSE. Metrics like MSE, IoU, DCS provide a good look into the properties of the representational space, but lacks link to the practical applicability of iSAEs, e.g. identifiability of meaningful concepts.
- **Interpretability and control**: in the discussion, the example provided for steering conflates interpretability and control, it is clear from recent works that clear interpretability is not always required for control of model behaviors.
- **Minor**: some notational clarity can be improved, e.g. the definition of parameters c in equation (2); parameter d in Section 3.1 bidirectional features

---

> ### Author Rebuttal · Authors · 2026-03-31
>
> Thank you for your kind words and attention to detail. We aim to address your weaknesses and key questions point by point.
>
> **(W1) Identifiability definition**
>
> We have now formally defined SAE identifiability separately from our theorem:
>
> > Mechanistic interpretability is often framed as recovering the “true” data-generating concepts under the linear representation hypothesis. However, without access to the true concepts, this form of identifiability is impossible to evaluate. However, the run-to-run stability of the learned concepts (or atoms, in the language of dictionary learning) and sparse codes, is more easily assessed (by varying the training seed), and is a pre-requisite for stable recovery of the true concepts often aimed for in mechanistic interpretability and other SAE applications. As such, we adopt the following definition of identifiability for SAEs.
> >
> > *Definition.* Let $P(x)$ denote a data distribution supported on $\mathcal{X} \subseteq \mathbb{R}^N$. Let $f, f’ : \mathcal{X} \rightarrow \mathcal{Z} \subseteq \mathbb{R}^K$ denote the encoders of two $k$-sparse SAEs trained independently on $P(x)$ by minimizing the expected loss $E_x[\mathcal{L}_\theta(x)]$, and let $D, D’ \in \mathbb{R}^{N \times K}$ denote their decoders. Then, the SAE model is $\epsilon$-nearly identifiable in the limit of infinite data if $||f(x) - \Pi f’(x)||_2 \leq \epsilon_z$ almost everywhere and $||D - D’ \Pi||_2 \leq \epsilon_D$ for $\epsilon_z, \epsilon_D > 0$ and some signed permutation matrix $\Pi \in \mathbb{R}^{K \times K}$.
> >
> > Intuitively, this definition says that an SAE is identifiable if independent trainings of the model (on infinite data) are guaranteed to yield the same solution, up to some trivial equivariances in the model, namely the ordering and sign of the atoms in the dictionary (and therefore the sparse codes).
>
> **(W2) Clarity of assumptions**
>
> We will move the assumptions block from the appendix to the main body. We have also added a Figure 1, and in the caption have enumerated the key assumptions that enable our proof strategy.
>
> **(W3/4/5) Empirical evaluation/ablations**
>
> We have added experiments on patch tokens from *DINOv2*. The performance of the model in this setting is very similar to Pythia160M. Additionally, we found that reconstruction performance is improved in the baseline TopK model by using new AuxK loss hyperparameters (how long before a token is considered dead). This improvement carries over to Pythia-160M as well, and so will update these results accordingly. Importantly, iSAE performs comparably in reconstruction and much better in identifiability still, and iSAE-ME performs markedly better in reconstruction. We have updated the discussion accordingly, with attention to the fact that these results are somewhat discordant with the findings in Zhu et al. (ICLR 2026).
>
> We additionally ran SAEBench (core and probing subtasks) on all models we report on in our paper for Pythia160M. The results suggest that all AbsTopK models generalize to new datasets better than the TopK baseline. Additionally, iSAE models have better held-out cross-entropy loss reconstruction performance and perform comparably or better on sparse probing tasks.
>
> **(W6) Steering**
>
> Thank you for your comment. You are correct to point out that data-driven approaches to steering might not require interpretability. The updated definition of identifiability and improved following paragraph hopefully makes this clearer.
>
> > Sparse autoencoders are increasingly being used to interpret and interact with the large-scale neural networks used in practice, such as large language models. In this setting, identifiability of the sparse autoencoder is a “bare minimum” requirement for certain applications. For example, mechanistic interpretability (MI) aims to uncover the ``true'' hidden workings of the model, assuming that there is a “true mechanism”. So, if two trained SAEs uncover two different candidate mechanisms in the form of concepts, they surely cannot both be correct. Furthermore, many approaches for model oversight (Li et al., 2026) and model steering (O’Brien et al., 2025) rely on concept identification, rendering SAE identifiability paramount (Cywinski et al., 2025).
>
> **(KQ1/2/3) Concept distribution / SAEs per variant / Permutations**
>
> At this time, we have only computed the “dead concept” rate, which is zero or near-zero in all models in all settings, due to the employed auxiliary loss. We fit three SAEs per variant, giving three pairs. We have updated the explanation to clarify that the permutation is simply a reordering of the SAEs neurons.
>
> **References**
>
> 1. Zhu et al. AbsTopK: Rethinking Sparse Autoencoders For Bidirectional Features, ICLR 2026.
> 2. Cywinski et al. SAeUron: Interpretable Concept Unlearning in Diffusion Models with Sparse Autoencoders, ICML 2025.

---

> > ### Author Rebuttal · Reviewer_m835 · 2026-04-03
> >
> > Thank you for the rebuttal, my concerns are mostly addressed.

---

### Official Review · Reviewer_VvMk · 2026-03-13

**Soundness:** 1
**Presentation:** 1
**Significance:** 2
**Originality:** 2
**Overall Recommendation:** 1
**Confidence:** 3

**Summary:**

This paper studies SAEs (sparse autoencoders) and presents some analyses on their theoretical properties. The paper introduces a new definition, aRIP, which is related to the RIP in compressive sensing, and develops a few experiments for their new approach. Experimental design shows performance gains of their approach over the classical ones.

**Compliance With Llm Reviewing Policy:**

Affirmed.

**Final Justification:**

I keep my assessment; the current manuscript does not meet the standard on presentation.

**Key Questions For Authors:**

1. Does iSAE mean "iterated SAE"? I could not find it in the paper.
2. Is eq. (10) the key design for iSAE?
3. The paper assumes SAE takes the form (1). Does it imply that z is generated by only an activated linear layer?

**Limitations:**

Yes

**Strengths And Weaknesses:**

## Soundness
Due to the presentation issues (see below), it's hard to assess the correctness of the paper's technical content. However, some obvious issues include
- Definition 3.2: Does (5) hold almost surely or in probability? It is not clear what the relationship is between $Z$ and $z$, or what the role of $P(z)$ is.
- Table 1: It could be interesting to discuss why iSAE-ME achieves almost perfect performance.
## Presentation
- The presentation is so poor that a reader cannot identify where iSAE, the proposed method, is formally defined in the paper.
- I don't think Theorem 3.1 in the manuscript is a valid mathematical statement in the current form. On top of that, it is referred to as "the following lemma" in the main text (right before Theorem 3.1), and is referred to as Theorem A.1 in the appendix.
- Other typos, e.g., the $\mathbf{c}$ in Eq. (2).
- Undefined notations, e.g., $z_S$ in eq. (5).
- Captions of all Tables shall be rewritten to better convey the message. The current ones are not formal sentences.
- Section 5, Discussion is long but not informative.
## Significance
Given SAE's broad applications, the study of SAE is a significant topic. However, the paper does not help provide a better understanding of the topic.
## Originality
The connection to compressive sensing, e.g., RIP, might be interesting, though their proposed aRIP seems to be ill-defined.

---

> ### Author Rebuttal · Authors · 2026-03-31
>
> Thank you for your review. We are sorry you had difficulty following the paper. We acknowledge a handful of typos, which have been corrected:
>
> * Equation (1) discussion had incorrect dimensionalities for the bias terms.
> * Equation (2) incorrectly replaced $\mathbf{z}$ with $\mathbf{c}$.
> * Equation (4)/(5) dropped an exponent.
>
> However, we note that at least one reviewer praised the clarity of our presentation. We do appreciate that some changes will be required to clarify our findings for some readers.
>
> To that end, we aim to discuss some of your other concerns.
>
> **(S1) Definition 3.2 and Theorem 3.1**
>
> As stated in the definition, the statement must hold for every sparse support $S$ in the support of the probability distribution over sparse supports (this is an *almost sure* statement, it is not clear what you mean by *in probability* – there is no notion of convergence here). Recognizing that this vocabulary might be unclear (because the word support is used to refer to both sparse supports and the support of a probability distribution), we will clarify the definition further.
>
> Without further description, it is not clear why you feel Theorem 3.1 is not mathematically valid. We will adjust the formatting so that it is more in keeping with the other Theorem in the text. The approach of Theorem 3.1 is fairly straightforward counterexample construction: problematic dictionaries exist, and if an SAE “lands” on such a problematic dictionary to approximate some distribution $P(x)$, the sparse codes are not identified (i.e. might be inconsistent, even if the dictionary is consistently learned). In response to your feedback and from another reviewer to reduce the included narration, we have updated the Theorem as follows:
>
> > Theorem. There exists a dictionary $D$ such that even if a continuous sparse encoder $f$, and error tolerance $\epsilon \geq 0$ exist such that the reconstruction error is bounded $||D f(x) - x|| \leq \epsilon$ almost everywhere with respect to a distribution over inputs $P(x)$ and all concepts are activated with positive probability under $P(x)$, there exists some other continuous encoder $f’$ that reconstructs equally well yet $f(x) \neq f’(x)$ (and furthermore the sparsity pattern in $f(x)$ and $f’(x)$ differ) with positive probability under $P(x)$.
>
> **(S2) Table 1: iSAE-ME near-perfect performance**
>
> The following existing explanation in Section 4.3 directly addresses this concern:
>
> > In the synthetic regime, adding the multistep encoder is crucial to realizing the identifiability gains from the improved dictionary conditioning in the SAE. In particular, even when bidirectional features and dictionary conditioning (previous sections) are enough to improve the quality of the dictionary and therefore the oracle solver, the default near-linear encoder cannot properly amortize codes near the true solution. On the other hand, iSAE-ME learns a nearly perfectly stable model in the synthetic regime as a result of its improved expressivity.
>
> Do you have specific concerns with this explanation?
>
> **(KQ1) iSAE**
>
> iSAE stands for “identifiable SAE”. We have clarified this in several parts of the paper.
>
> **(KQ2) iSAE design**
>
> No. As discussed in Section 4.2, iSAE refers to the regularization term (7). As discussed in Section 4.3, iSAE-ME refers to the combination of the regularization term (7) and multistep iterated encoding (10).
>
> **(KQ3) Equation (1)**
>
> Thank you for this question. Yes, all common SAE variants that we are aware of use an activated linear layer as the encoder. In particular, the TopK baseline used in this paper uses a model exactly of the form of equation (1), upon which our two variants are based. To our knowledge, other encoders have been experimented with (for example MLPs in the popular `overcomplete` library), but in our experiments these did not yield substantially different results from the linear case, reflecting that the subset selection problem is “the hard part” of amortized sparse coding [1].
>
> [1] Gregor, K. and LeCun, Y. Learning fast approximations of sparse coding. ICML 2010.

---

> > ### Author Rebuttal · Reviewer_VvMk · 2026-04-03
> >
> > I have read all the reviews and rebuttals. I acknowledge the authors' effort in answering the questions. However, I do see many remaining issues, particularly in presentation, which make the current manuscript well below the standard for publication.
> >
> > For example, the authors mentioned
> > > iSAE stands for “identifiable SAE”. We have clarified this in several parts of the paper.
> >
> > I cannot see where the "several parts" are.
> >
> > As a side note, I do not see the value of putting "at least one reviewer praised the clarity of our presentation", as the review reflects my judgment.

---

> > > ### Author Response · Authors · 2026-04-05
> > >
> > > Thank you for your acknowledgement. Unfortunately, we are not permitted to update the PDF, so you are correct that you cannot see our edits in real time as with some other venues. The best we can do is commit to making the changes discussed here.

---

### Decision · Program_Chairs · 2026-04-30

**Decision:**

Accept (regular)

**Comment:**

This paper addresses an important weakness of sparse autoencoders: instability and non-identifiability across training runs. It connects SAEs to dictionary learning and compressed sensing, arguing that stable dictionaries alone are insufficient and that sparse codes must also be stable; to this end, it introduces a data-dependent relaxation of RIP, aRIP, together with iSAE, a modified SAE designed to improve identifiability and reconstruction. The reviews were overall more positive than negative: several reviewers found the theoretical direction interesting and potentially valuable, particularly the bridge to dictionary-learning theory and the emphasis on code consistency in addition to dictionary stability. The main concerns were about presentation, clarity, and somewhat limited empirical breadth, and the most negative review appears to have been driven largely by these exposition issues rather than by a substantive disagreement with the paper’s core contribution. The rebuttal was helpful and clarified many points, especially around the definition of identifiability, the role of iSAE, and the assumptions behind the theory. Overall, the paper was seen as technically solid and worth building on, though its impact was partly obscured by the clarity of the submitted draft.